# Splicing factor SRSF1 deficiency in the liver triggers NASH-like pathology and cell death

Waqar Arif [1,2], Bhoomika Mathur [3], Michael F. Saikali[4], Ullas V. Chembazhi [1], Katelyn Toohill[1], You Jin Song[5], Qinyu Hao[5], Saman Karimi[6], Steven M. Blue [7,8], Brian A. Yee [7,8], Eric L. Van Nostrand [7,8,11], Sushant Bangru[1,9], Grace Guzman [6], Gene W. Yeo [7,8], Kannanganattu V. Prasanth [5,9], Sayeepriyadarshini Anakk[3,9], Carolyn L. Cummins [4] & Auinash Kalsotra [1,9,10] ✉

Regulation of RNA processing contributes profoundly to tissue development and physiology. Here, we report that serine-arginine-rich splicing factor 1 (SRSF1) is essential for hepatocyte function and survival. Although SRSF1 is mainly known for its many roles in mRNA metabolism, it is also crucial for maintaining genome stability. We show that acute liver damage in the setting of targeted SRSF1 deletion in mice is associated with the excessive formation of deleterious RNA–DNA hybrids (R-loops), which induce DNA damage. Combining hepatocyte-specific transcriptome, proteome, and RNA binding analyses, we demonstrate that widespread genotoxic stress following SRSF1 depletion results in global inhibition of mRNA transcription and protein synthesis, leading to impaired metabolism and trafficking of lipids. Lipid accumulation in SRSF1-deficient hepatocytes is followed by necroptotic cell death, inflammation, and fibrosis, resulting in NASH-like liver pathology. Importantly, SRSF1-depleted human liver cancer cells recapitulate this pathogenesis, illustrating a conserved and fundamental role for SRSF1 in preserving genome integrity and tissue homeostasis. Thus, our study uncovers how the accumulation of detrimental R-loops impedes hepatocellular gene expression, triggering metabolic derangements and liver damage.

The liver performs hundreds of diverse functions that include detoxifying blood of potentially harmful drugs, producing bile for fat absorption, and processing nutrients to meet metabolic demands during fasting and fed states[1,2]. Therefore, it is not surprising that dysfunction of the liver is associated with poor prognosis and high rates of mortality[3,4]. Besides viral infections, excessive alcohol consumption and

nonalcoholic fatty liver disease are the biggest risk factors for hepatic failure. Particularly, the prevalence of nonalcoholic fatty liver disease is rapidly increasing worldwide, posing a significant public health threat[5–7]. The disease starts with an excess buildup of fat within the liver, but as the condition worsens, the fat deposition (steatosis) is accompanied by severe inflammation (hepatitis) and scarring (fibrosis) that

[1]Department of Biochemistry, University of Illinois Urbana-Champaign, Urbana, IL, USA. [2]College of Medicine, University of Illinois at Urbana-Champaign, Urbana, IL, USA. [3]Department of Molecular and Integrative Physiology, University of Illinois Urbana-Champaign, Urbana, IL, USA. [4]Department of Pharmaceutical Sciences, Leslie Dan Faculty of Pharmacy, University of Toronto, Toronto, ON, Canada. [5]Department of Cell and Developmental Biology, University of Illinois Urbana-Champaign, Urbana, IL, USA. [6]Department of Pathology, College of Medicine, Cancer Center, University of Illinois Hospital and Health Science Chicago, Chicago, IL, USA. [7]Department of Cellular and Molecular Medicine, University of California at San Diego, La Jolla, CA, USA. [8]Institute for Genomic Medicine, University of California at San Diego, La Jolla, CA, USA. [9]Cancer Center @ Illinois, University of Illinois Urbana-Champaign, Urbana, IL, USA. [10]Carl R. Woese Institute of Genomic Biology, University of Illinois Urbana-Champaign, Urbana, IL, USA. [11]Present address: Department of Biochemistry and Molecular Biology, Baylor College of Medicine, Houston, TX, USA. ✉e-mail: kalsotra@illinois.edu

leads to cirrhosis and, finally, hepatocarcinogenesis[8]. The disease is then referred to as Non-Alcoholic Steatohepatitis (NASH).

Hepatocytes, which are highly differentiated and quiescent cells, are the primary functional units of the liver. While most gene expression studies focus on transcriptional control of hepatocyte function and development, several recent studies have highlighted critical roles for post-transcriptional gene regulatory mechanisms[9–14]. These mechanisms, which include mRNA capping, splicing, polyadenylation, and editing, are coordinated by a complex interplay between mRNA and RNA binding proteins (RBPs) and, in general, control the expression of transcripts by altering their sequence, stability, localization, or translation efficiency[15–17]. The conserved SR protein family, which includes 12 canonical members, are well-characterized RBPs that regulate multiple aspects of mRNA metabolism. They all share conserved modular structural elements consisting of either one or two N-terminal RNA recognition motifs, which define their RNA sequence binding specificity, and a C-terminal arginine- and serine-rich domain[18]. Because the genetic deletion of many SR proteins is embryonic lethal, their physiological roles in vivo are largely unexplored. However, the latest studies using conditional deletion models are now highlighting the importance of SR proteins in maintaining liver homeostasis. For example, hepatocyte-specific deletion of *Srsf2* and *Srsf3* in mice resulted in acute liver damage with defects in metabolic functions[19,20]. Likewise, *Srsf10* has also been implicated in liver dysfunction with reduced levels leading to increased hepatic lipogenesis and steatosis[21].

Of the SR protein family, SRSF1 (ASF/SF2) was the first to be identified and is considered the archetype member. Though initially characterized as a splicing factor, SRSF1 has since been found to regulate nearly every aspect of the mRNA lifecycle, including mRNA transcription, non-sense mediated decay (NMD), mRNA export, and translation[22–24]. In addition to mRNA metabolism, SRSF1 is also involved in other biological functions such as microRNA processing, maintaining genomic stability, and nucleolar stress[24,25]. The first glimpse of SRSF1's role in tissue physiology began with in vivo investigation in mice hearts. Cardiomyocyte-specific deletion of *Srsf1* in mice resulted in missplicing of the $Ca^{2+}$/calmodulin-dependent kinase IIδ, leading to severe defects in excitation-contraction coupling and heart failure[26]. Additional studies since then have shown SRSF1 involvement in T-cell differentiation, vascular smooth cell proliferation, and skeletal muscle development[27–31]. Furthermore, SRSF1 levels are elevated in many different cancers, and it is considered a proto-oncogene[32–34]. Indeed, orthotopic transplantation of mouse mammary epithelial cells overexpressing SRSF1 is sufficient to promote tumorigenesis[35,36]. While SRSF1 is well-studied in the context of cancer and cardiac physiology, its function in the liver is not well understood.

Here, we generated constitutive and inducible mice models of hepatocyte-specific *Srsf1* deletion to define its in vivo function in liver physiology. We found that SRSF1 inactivation is detrimental to cell viability and triggers NASH-like pathology. Mechanistically, we show that SRSF1-deficient hepatocytes accumulate deleterious RNA-DNA hybrids (R-loops) and develop extensive DNA damage. The overwhelming damage results in transcriptional inhibition, missplicing, diminished protein synthesis, metabolic insufficiency, and cell death, inducing a compensatory regenerative response that gradually repopulates the liver with SRSF1-expressing hepatocytes. Importantly, the transient knockdown of SRSF1 in human liver cancer cells recapitulates the molecular pathogenesis identified in the animal models. Thus, our findings uncover new interconnections between genome stability, hepatocellular gene expression, and intermediary metabolism.

## Results

### Hepatic knockout of SRSF1 triggers immediate repopulation with SRSF1-expressing hepatocytes

To investigate the role of SRSF1 in liver physiology, we disrupted its expression in hepatocytes by crossing *Srsf1*flox/flox with *AlbCre*

transgenic mice. Although hepatocyte-specific SRSF1 knockout (SRSF1 HKO) mice were born at the expected Mendelian ratio and survived to adulthood, their livers displayed gross morphological defects during the early postnatal stages. At postnatal day (PN) 10, SRSF1 HKO livers were highly pale and yellow in color, signifying severe fatty infiltration of the tissue, referred to as steatosis (Fig. 1a). SRSF1 HKO mice also appeared to have stunted growth which is evident from their decreased weight trend in comparison to littermate controls (Supplementary Fig. 1a, b). Histological analysis of the liver tissue sections at early postnatal timepoints showed that, up to PN6, both SRSF1 HKO and control mice had similar lipid content. However, beyond this time point, lipid accumulation persisted in SRSF1 HKO livers, whereas in control mice, the lipid content had diminished (Fig. 1b; Supplementary Table 1). Furthermore, hepatocyte damage and death were evident starting at PN6 and continued through later timepoints (Supplementary Fig. 1c). Surprisingly, despite the drastic liver phenotype, SRSF1 levels in the knockout model at PN10 displayed only a two-fold reduction (Supplementary Fig. 1b). This finding was unexpected because previous reports utilizing a Cre-dependent reporter have shown uniform *AlbCre* transgene activity across all hepatocytes by PN3[37,38]. Hence, a greater reduction of SRSF1 was anticipated in the knockout model.

To determine the basis of this discrepancy, we performed immunofluorescent co-staining for SRSF1 and the hepatocyte marker HNF4α on liver sections at early postnatal timepoints (Fig. 1b). The greatest decrease in SRSF1 expression within hepatocytes was observed at PN6, which coincided with the onset of damage and steatosis detected histologically. After this time point, SRSF1-deficient hepatocytes were slowly replaced by an expanding population of SRSF1-positive hepatocytes (Fig. 1c). We hypothesized that the parenchymal repopulation with SRSF1-expressing hepatocytes resulted either from the expansion of wildtype hepatocytes that have escaped Cre-mediated SRSF1 knockout and/or continuous transdifferentiation of biliary epithelial cells (BEC) into hepatocytes. Irrespective of the mechanism, an increase in hepatic proliferation would be required to sustain the steady repopulation. As predicted, increased immunostaining of hepatocyte nuclei with Ki67, a marker of cell proliferation, was observed in SRSF1 HKO mice at PN10 (Supplementary Fig. 1c, d). While increased proliferation at early stages is expected, later-stage phenotypes are anticipated to differ depending on the mechanism of hepatocyte repopulation. For instance, if hepatocytes were escaping Cre-mediated knockout of *SRSF1*, the livers would repopulate with SRSF1-expressing hepatocytes, leading to normalization of liver function over time. Conversely, if transdifferentiation of BECs to hepatocytes was the primary mechanism, chronic injury and eventual liver failure would be likely as transdifferentiated hepatocytes would cycle through continuous cell death.

To determine the primary mechanism of hepatocyte repopulation in SRSF1 HKO livers, we performed immunofluorescence and RT-qPCR analysis at later time points. In agreement with the first mechanism, quantification of SRSF1-expressing hepatocytes at 1- and 3 months revealed a steady repopulation of the liver parenchyma (Fig. 2a). Specifically, at 1 month, the SRSF1 HKO livers were nearly 60% repopulated with SRSF1-expressing hepatocytes, and by 3 months, the repopulation was essentially complete. This finding correlated well with the gradual reversal of pathological changes detected in juvenile SRSF1 HKO mice (Supplementary Fig. 2a–d). For instance, SRSF1 HKO had stunted growth with lower total body and adipose weights at 1 and 3 months. However, by 6 months, these differences normalized to littermate controls as liver function was restored.

The phenomenon of liver repopulation with wildtype hepatocytes in a hepatocyte-specific conditional knockout mouse model has been previously reported. Sekine et al. found that hepatocyte-specific deletion of *Dicer1* results in overwhelming cell death

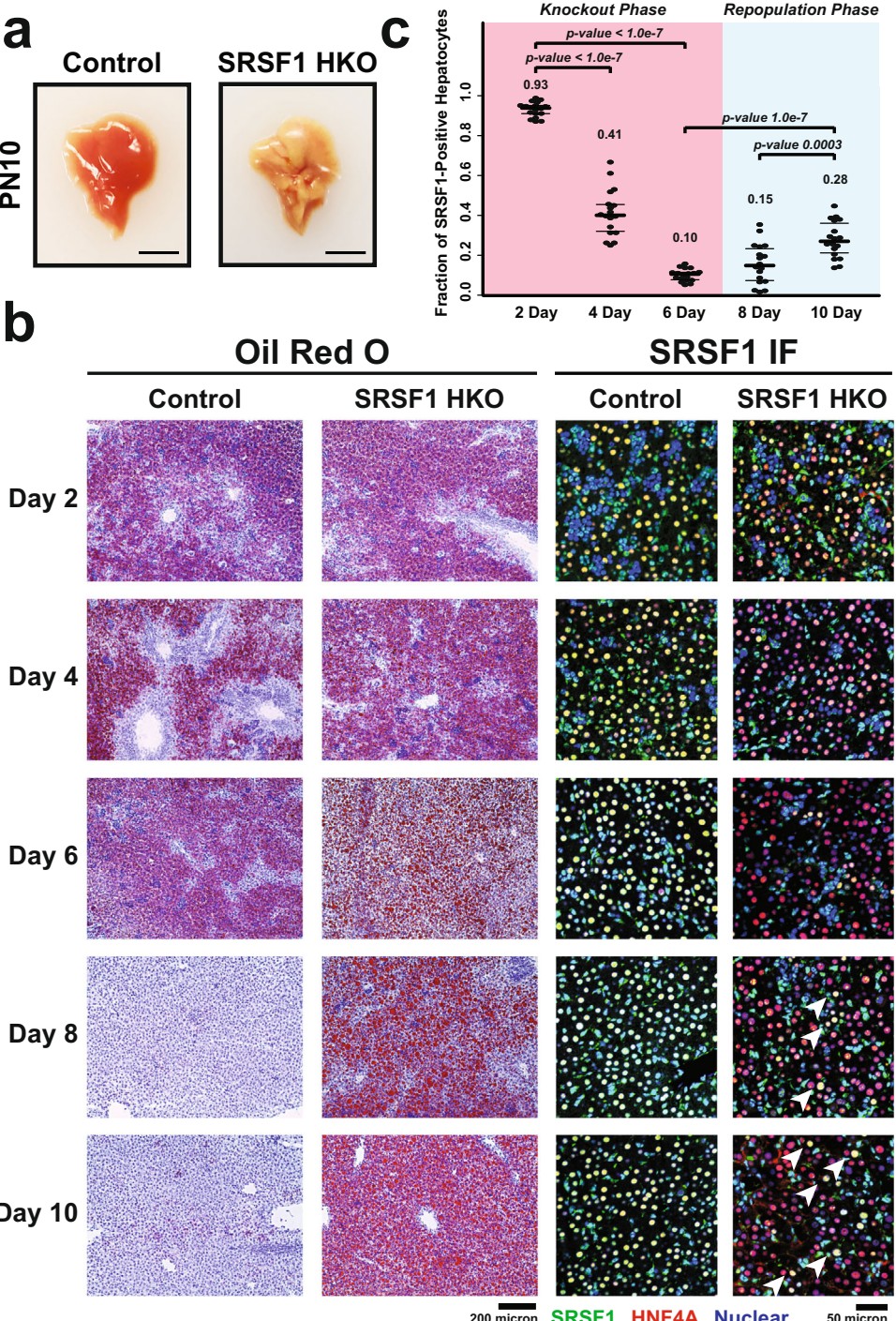

**Fig. 1 | Knockout of SRSF1 in hepatocytes triggers a regenerative response.**
**a** Representative gross images of livers harvested from Control (*AlbCre*[+/−]) and
SRSF1 HKO (*SRSF1*[*flox/flox*]; *AlbCre*[+/−]) mice 10 days after birth (*n* = 6 biologically
independent animals per group). Scale bar = 1 cm. **b** Representative histological
(Oil Red O) and immunofluorescence (IF) images of liver sections from Control and
SRSF1 HKO mice at indicated timepoints (*n* = 3 biologically independent animals
per group, 6 fields per replicate). IF images show co-staining with SRSF1 (green),
HNF4α (red), and Nuclei (blue). White arrows indicate the repopulation of tissue
with SRSF1-expressing hepatocytes. **c** Quantification of the fraction of SRSF1-
expressing hepatocytes per field from IF images at each timepoint (*n* = 3 biologi-
cally independent animals, 6 fields per replicate). Values are displayed as mean ±
SD. A two-way ANOVA analysis was performed with Tukey multiple comparisons of
means. Source data are provided as a Source Data file.

followed by regeneration and progressive repopulation with
DICER1-expressing hepatocytes[39]. Further, they demonstrated that
the repopulating hepatocytes had escaped *Dicer1* knockout by
silencing the expression of the *Cre* transgene. Therefore, to assess if
*Cre* silencing was occurring in SRSF1 HKO, the abundance of *Cre*
mRNA was measured using qRT-PCR at PN4, 1-month, and 3-month

time points (Fig. 2b). Notably, compared to controls, the SRSF1 HKO
livers showed a striking downregulation of *Cre* expression
over time, thus allowing *SRSF1* to remain intact. These data provide
strong evidence for the progressive repopulation of SRSF1
HKO livers with hepatocytes that have escaped Cre-mediated
recombination.

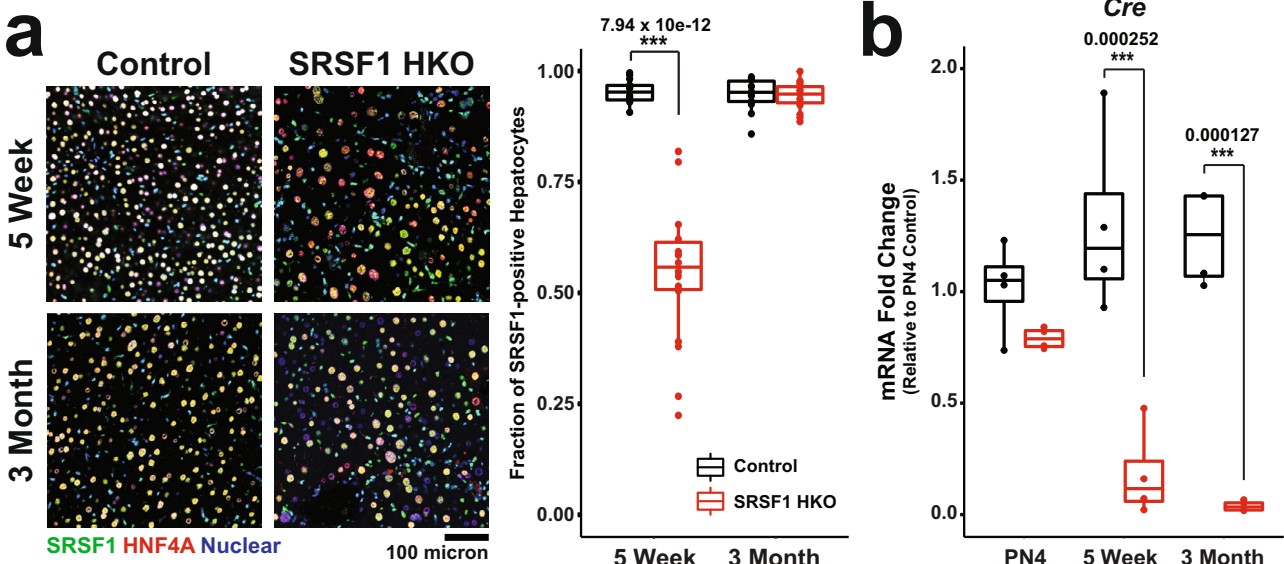

**Fig. 2 | Hepatocytes in SRSF1 HKO mice eventually circumvent knockout via AlbCre repression. a** *Left:* Representative IF images of liver sections from Control and SRSF1 HKO mice at indicated timepoints ($n = 3$ biologically independent animals, 6 fields per replicate). IF images show co-staining with SRSF1 (green), HNF4α (red), and Nuclei (blue). *Right:* Quantification of the fraction of SRSF1-expressing hepatocytes per field from IF images at each timepoint. **b** Relative mRNA expression (qPCR) of AlbCre normalized to *36B4* expression in Control and SRSF1 HKO mice ($n = 4$ mice per group) at the indicated timepoints. PN4, post-natal day 4. Two-way ANOVA analysis with multiple testing correction was performed to determine the significance between 2 groups at 2 timepoints. Significant differences are designated by "***" with the p-value listed above. Source data are provided as a Source Data file.

## SRSF1 HKO mice exhibit acute and reversible NASH-like liver injury

Although SRSF1 HKO livers began to overcome Cre-mediated knockout early in postnatal development, these mice still displayed severe liver dysfunctions at the adult stages. Histological analysis of SRSF1 HKO livers at 1-, 3- and 6-month time points presented pathology comparable to the human disease known as Nonalcoholic steatohepatitis, or NASH (Fig. 3; Supplementary Table 2). Key hallmarks of NASH include the progressive development of steatosis, ballooning degeneration, necroinflammation, and perisinusoidal fibrosis[40,41]. At 1-month, widespread injury was detected with necrotic and ballooning hepatocytes accompanied by infiltrating inflammatory cells (Fig. 3). However, the damage and inflammation were drastically reduced at later time points (Supplementary Table 2). Similarly, significant lipid accumulation was observed at 1 and 3 months, but by 6 months, the steatosis was strikingly diminished (Fig. 3). A defining feature of NASH is scar tissue formation, or fibrosis, as the liver attempts to repair the injured tissue[40]. As anticipated, fibrosis was detected in SRSF1 HKO mice, with peak fibrosis seen at 3 months and slight bridging fibrosis at 6 months (Fig. 3). But, unlike NASH, where damage worsens with time, SRSF1 HKO liver injury subsided with age and, except for the intrahepatic fibrosis, the mice recovered as *SRSF1*-expressing hepatocytes repopulated the liver (Fig. 3; Supplementary Table 2).

Consistent with the histological findings, liver function and metabolic tests in SRSF1 HKO indicated chronic injury with eventual normalization (Supplementary Table 3; Supplementary Fig. 2e). For instance, serum ALT and AST levels were drastically elevated at 1 month, signifying liver damage, but the levels gradually decreased with age and by 6 months were indistinguishable from controls. On the other hand, serum triglycerides and cholesterol levels showed the opposite trend with decreased levels at early ages, which increased to baseline by 6 months. This is expected since the liver is involved with lipid trafficking and cholesterol synthesis. Despite abnormal liver function tests, glucose metabolism remained intact in SRSF1 HKO, indicated by their normal fasting glucose levels and glucose tolerance

tests (Supplementary Table 3 and Supplementary Fig. 2e). Altogether, these findings demonstrate that liver damage in SRSF1 HKO mice occurs in two phases consisting of 1) acute injury which begins immediately after birth and 2) chronic injury that lasts about three months, at which point the liver is completely repopulated with *SRSF1*-expressing hepatocytes.

## Hepatic ablation of SRSF1 induces inflammatory and fibrotic gene signatures

To study transcriptome changes in SRSF1 HKO during both early and late phases of injury, we performed RNA sequencing (RNA-seq) on isolated hepatocytes from PN10 (early) and 1-month (late) old mice. Differential gene expression analysis using DESeq2 showed drastic changes in mRNA abundances ($|\log_2\text{FoldChange}| \geq 1$, FDR < 0.10) at both early and late timepoints with approximately 8.6% and 10.0% of expressed hepatic genes affected, corresponding to 1825 and 1,902 transcripts, respectively (Fig. 4a). Furthermore, a larger fraction of differentially expressed genes (DEGs) increased in expression, with ~63% at the early and ~79% at the late phases respectively. DEGs shared between the early and late time points encompassed 544 genes corresponding to about one-third of both gene sets. Although the expression of the overlapping gene set had a strong linear correlation ($R^2_{\text{pearson}} = 0.647$), the degree of fold change tended to be greater during the late phase (Fig. 4b). Hence, genes that were upregulated early were further potentiated during the late phase.

As substantial alterations to the transcriptome were triggered in response to SRSF1 deletion, we hypothesized these changes are induced to mitigate the hepatic insufficiency and injury resulting from the loss of SRSF1 activity. To identify biological processes associated with the transcriptome changes, gene ontology enrichment analysis was performed on DEGs in SRSF1 HKO (Fig. 4c). Genes upregulated during the acute phase in SRSF1 HKO were highly enriched for inflammatory processes such as chemokine signaling and leukocyte migration. This is expected as widespread cell death frequently triggers inflammation to promote cellular recruitment for tissue repair. Indeed, infiltration of inflammatory cells into the tissue parenchyma

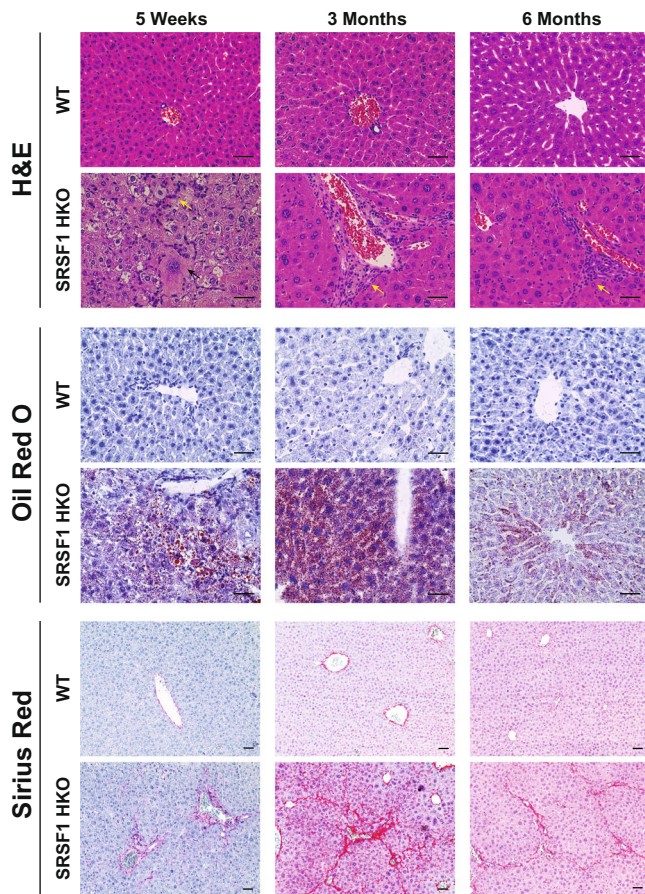

**Fig. 3 | Acute and reversible NASH-like pathology in adult SRSF1 HKO mice.**
Representative images of H&E, Oil Red O (red, neutral lipids), and Sirius Red (red, collagen) histological staining of liver tissue harvested from wildtype (WT) and SRSF1 HKO mice at the indicated ages (*n* = 6 biologically independent animals per group). Yellow and black arrows in H&E staining indicate inflammatory infiltration and ballooning degeneration, respectively. Scale bar = 100 microns. Source data are provided as a Source Data file.

was seen in SRSF1 HKO at the late time point. However, the inflammation resolved as the liver repopulated with SRSF1-expressing hepatocytes, resulting in decreased necrosis and wound healing. This response pattern was further corroborated by the expression profiles of various inflammatory and fibrosis markers (Fig. 4d). Of note, genes upregulated at the later time point were primarily involved in cell division and mitotic processes, which points to the compensatory regenerative response occurring in the SRSF1 HKO livers (Fig. 4c). Conversely, transcripts downregulated at either time point in SRSF1 HKO were associated with key metabolic pathways. During the early phase, pathways involved in sterol transport and fatty acid oxidation were downregulated, whereas the late phase encompassed the downregulation of sulfur amino acid, ornithine, and urea metabolism. In general, this suggests that SRSF1 HKO livers exhibit reduced metabolic capacity (Fig. 4c). Collectively, our data show that hepatic loss of SRSF1 triggers a robust expression of transcripts to facilitate inflammation, which then gradually transitions to proliferative and wound healing processes as the liver regenerates.

Considering that SRSF1 is a canonical splicing factor, loss of its activity is anticipated to result in extensive splicing defects. Using rMATS, a computational tool for quantifying differential splicing from RNA-seq, we identified 852 and 765 significantly changing splicing events (ΔPSI > 15%; FDR < 0.10) at the early and late phases, respectively (Supplementary Fig. 3a, b). The transcripts with differentially spliced exons (DSE) were enriched in a wide variety of functions

ranging from metabolic processes to chromosomal organization (Fig. 4c). However, due to the acute injury and regenerative response in this model, the interpretation of these findings is complicated and likely reflects secondary changes to liver injury and damage. Therefore, to identify the molecular events leading to damage upon SRSF1 ablation, a model was required which would allow capturing of SRSF1-deficient hepatocytes before the onset of the injury.

## Acute hepatic knockout of SRSF1 in adult mice recapitulates SRSF1 HKO pathology

We developed an additional model which allows for acute knockout of SRSF1 in hepatocytes of adult mice (acSRSF1 HKO). This was achieved using adeno-associated viral vectors expressing the Cre recombinase driven by the hepatocyte-specific thyroxine-binding globulin (TBG) promoter (Fig. 5a). A benefit of this model is that it allows the study of primary molecular changes resulting from SRSF1 depletion by permitting the isolation of SRSF1-deficient hepatocytes before the development of pathology. Robust depletion of SRSF1 protein was achieved in the livers of adult mice after 2 weeks of AAV8-TBG-iCre viral vector transduction (Fig. 5b). Prior to performing in-depth molecular studies on acSRSF1 HKO mice, we verified if this model develops liver pathology like SRSF1 HKO mice. Histological staining revealed mild microsteatosis at the 2-week time point with lipid droplets distributed near the central vein hepatocytes. Importantly, there was no evidence of damage or fibrosis at this time point (Supplementary Fig. 4a). By 4 weeks, acSRSF1 HKO livers displayed severe lipid accumulation with macrosteatosis throughout the tissue parenchyma. In addition, signs of cell death with ballooning degenerating hepatocytes were noted. To verify if the dying hepatocytes were initiating apoptosis, we performed TUNEL staining on liver tissue sections (Supplementary Fig. 4b). The staining showed no positive cells, suggesting that SRSF1-deficient hepatocytes were likely undergoing necrotic cell death. Despite the severe steatosis and damage by 4 weeks, acSRSF1 HKO mice did not exhibit any overt liver inflammation or fibrosis. These findings demonstrate that acSRSF1 HKO mice develop severe steatosis followed by necrosis, recapitulating the pathological progression seen in SRSF1 HKO mice.

A limitation of the SRSF1 HKO model was the early regenerative response and repopulation with SRSF1-expressing hepatocytes. While western blot analysis of acSRSF1 HKO livers showed efficient knockout of SRSF1 at 2 weeks, like the SRSF1 HKO model, re-expression of SRSF1 was noted by 4 weeks (Fig. 5b). To validate that this re-expression was due to the repopulation of the liver with SRSF1-expressing hepatocytes, immunofluorescence co-staining was performed on tissue sections (Fig. 5c). Indeed, the staining showed the reappearance of SRSF1-positive hepatocytes. Nonetheless, despite this repopulation, the acSRSF1 HKO model provided the opportunity to study SRSF1-deficient hepatocytes before the onset of damage.

## AcSRSF1 HKO mice develop acute hepatic damage

Next, we sought to understand the physiological changes occurring in the acSRSF1 HKO mice. We began by assessing changes in total body, liver, and adipose tissue mass following viral transduction (Fig. 5d). We found that acSRSF1 HKO mice maintained their body weight for up to 2 weeks, which then fell dramatically by 4 weeks. On average, acSRSF1 HKO mice lost ~15% of their starting total body weight, whereas control mice gained ~5% after 4 weeks. Regarding the liver and adipose tissue, we found that their masses trended reciprocally to each other (Fig. 5e, f). In agreement with the observed steatosis and hepatocyte dysfunction, liver mass steadily increased with time while adipose stores diminished. Given the notable steatosis, we wondered if the accumulating lipids primarily consisted of triglycerides or cholesterol (Fig. 6a). Therefore, total hepatic lipids were extracted from whole liver tissue, followed by quantification using colorimetric assays. Our measurements showed progressively elevated levels of hepatic triglycerides at

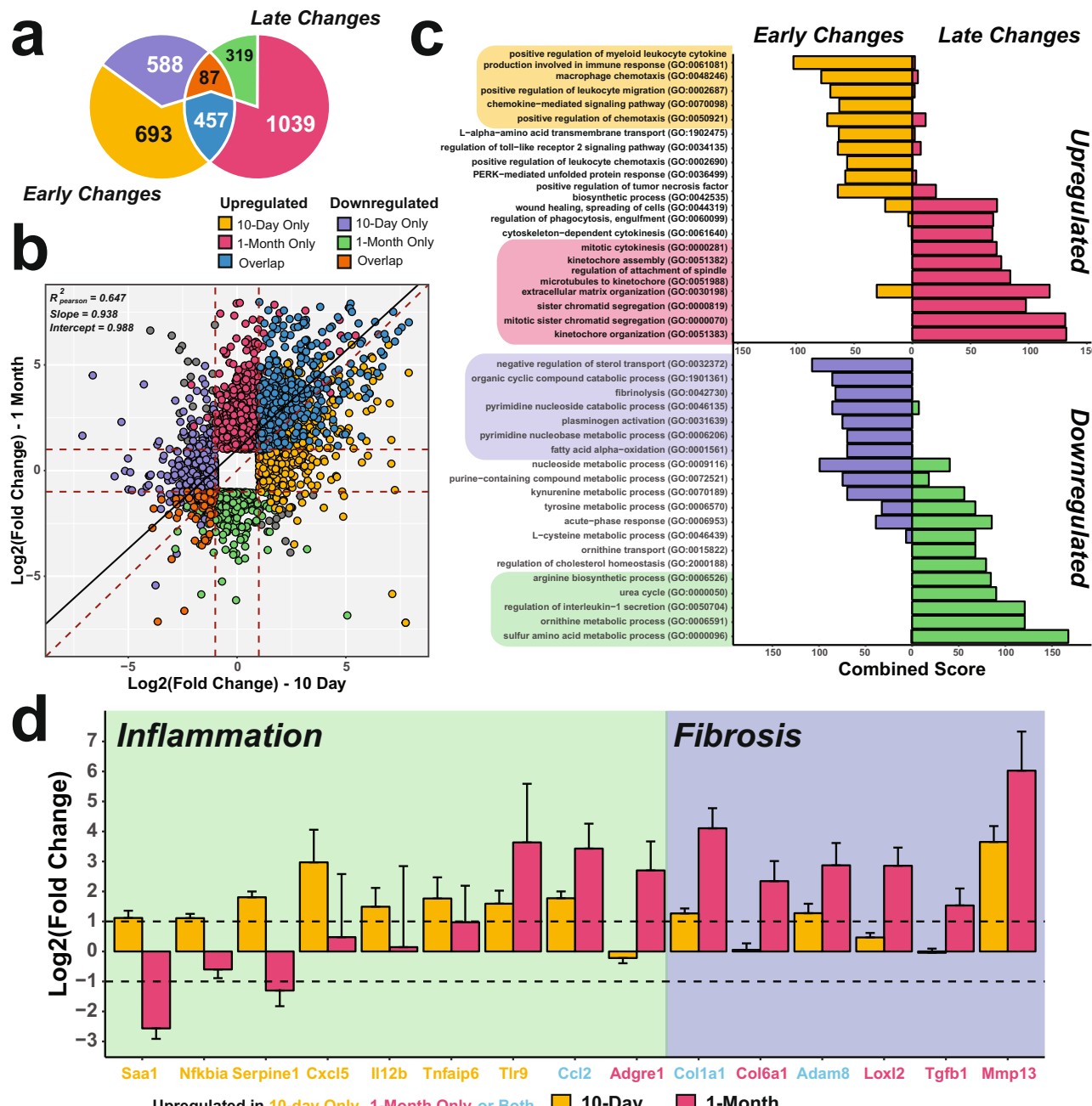

**Fig. 4 | Gene expression signatures in SRSF1 HKO hepatocytes transition from an early inflammatory to a late fibrotic phase. a** Overlap of differentially expressed genes from RNA-seq (FDR < 0.05, Wald test as described DESeq2; | Log₂(Fold Change)| ≥ 1) in 10-day (early) and 1-month (late) old SRSF1 HKO hepatocytes (n = 2 samples/condition). **b** Scatter plot showing the distribution of mRNA abundance fold changes in SRSF1 HKO with respect to controls at both early and late timepoints. **c** Gene ontology (GO) terms of upregulated and downregulated genes in SRSF1 HKO hepatocytes at 10-day and 1 month or early and late, respectively. **d** Bar plot of fold change values determined from RNA-seq analysis for genes involved in inflammation and fibrosis at both early (yellow bars) and late (red bars) timepoints (n = 2 samples per group per timepoint). The error bars represent the associated standard error estimated by DESeq2. The color of gene labels signifies significant upregulation in mRNA abundance at 10-day only (yellow), 1-month only (red), or both (blue). Source data are provided as a Source Data file.

2- and 4-weeks in acSRSF1 HKO mice and simultaneous reductions in cholesterol levels (Fig. 6b). To further evaluate the hepatic dysfunction in acSRSF1 HKO mice, biochemical profiling of the serum was performed (Fig. 6c). Measurement of serum ALT and AST activity, markers of liver injury, showed no significant difference at the 2-week time point between acSRSF1 HKO and control mice. However, by 4 weeks, their levels were strikingly elevated, signifying severe liver damage. This was also evident by the golden yellow appearance of the serum due to elevated bilirubin levels resulting from decreased clearance by the liver (Supplementary Fig. 4c).

Importantly, acSRSF1 HKO mice did not exhibit significant differences at 2 weeks for any of the measured serum parameters (Fig. 6c). These findings further support that this time point precedes any detectable damage and secondary effects. On the other hand, serum profiling of 4-week acSRSF1 HKO mice serum showed severe metabolic derangements. For instance, fasting glucose levels in acSRSF1 HKO were lower than normal, with an average concentration of 65 mg/dL at 4 weeks, signifying impaired gluconeogenesis. In a fasted state, the liver generates ketone bodies as an alternative energy source, reflected by their elevated serum levels. However, we noticed

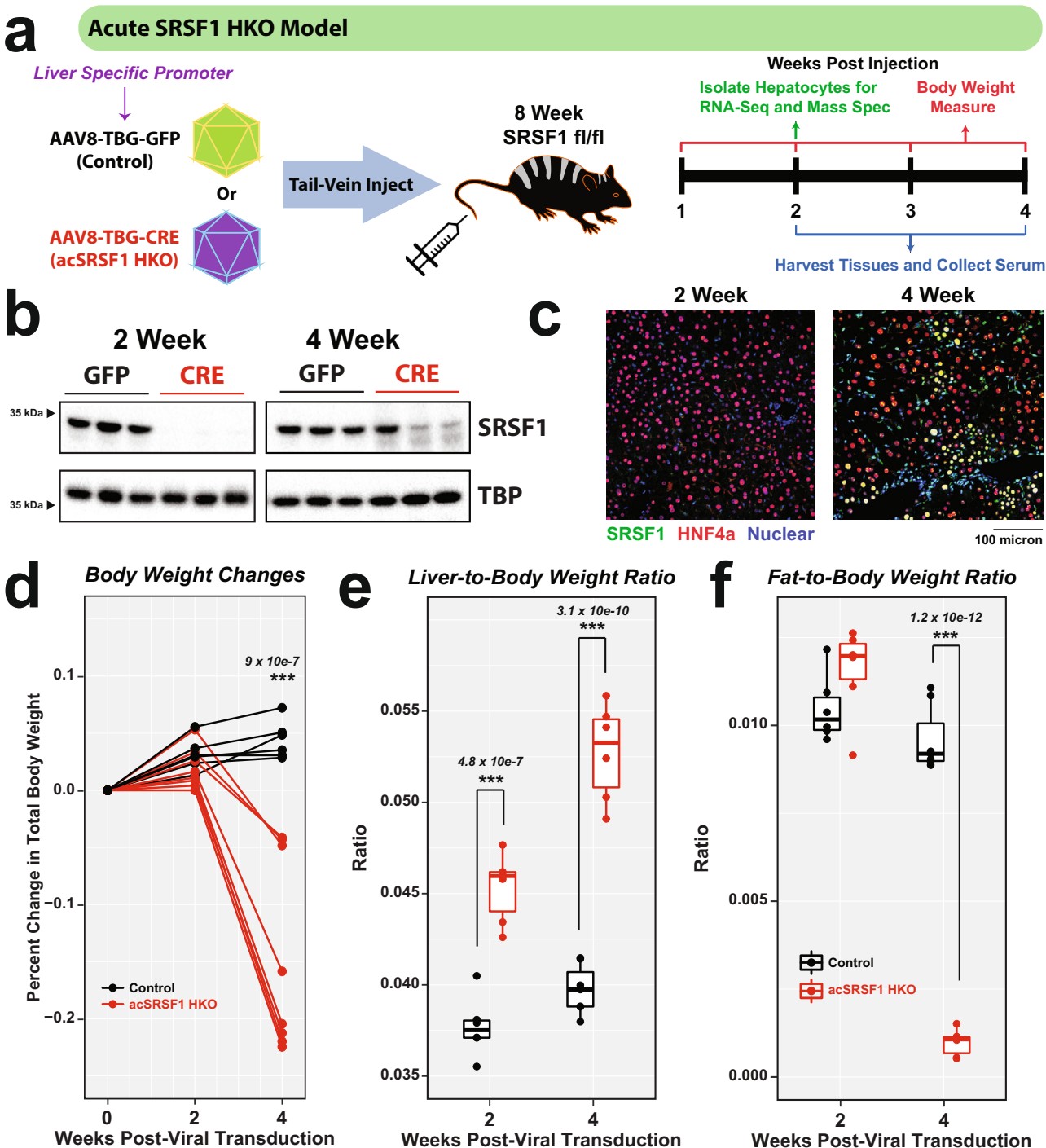

**Fig. 5 | Generation and characterization of mice with acute hepatocyte-specific knockout of SRSF1. a** Schematic description of acute hepatocyte-specific SRSF1 knockout (acSRSF1 HKO) mice model and experimental design. Transgene expression is driven by the liver-specific thyroxine-binding globulin (TBG) promoter. **b** Western blot showing hepatic SRSF1 protein levels in acSRSF1 HKO (CRE) relative to controls (GFP) at the indicated timepoints post viral transduction. TBP served as a loading control. **c** Representative IF images of liver sections from acSRSF1 HKO mice at indicated timepoints (*n* = 5 biologically independent animals, 6 fields per replicate). IF images show co-staining with SRSF1 (green), HNF4α (red), and Nuclei (blue). Yellow nuclei indicate SRSF1-positive hepatocytes. **d** The plot of percent change in total body weight at indicated timepoints. Weight before viral transduction is considered baseline (*n* = 6 and 8 biologically independent animals for control and acSRSF1 HKO, respectively). Box plots of (**e**) liver-to-body weight and (**f**) gonadal fat-to-body weight ratio in control and acSRSF1 HKO (*n* = 6 biologically independent animals per group per timepoint). Two-way ANOVA analysis with multiple testing corrections was performed to determine the significance between 2 groups at 2 timepoints. Significant differences are designated by "***" with the p-value listed above. Source data are provided as a Source Data file.

that ketone bodies in acSRSF1 HKO were strikingly diminished at 4 weeks despite being in a fasted state. In agreement with the decreased cholesterol in hepatic tissue, acSRSF1 HKO also showed declining serum cholesterol levels. This is expected with widespread liver damage since cholesterol is predominantly synthesized by hepatocytes. Likewise, fractionated serum profiles of 4-week acSRSF1 HKO mice showed marked depletion of lipoprotein particles that are synthesized and secreted by hepatocytes for systemic transport of

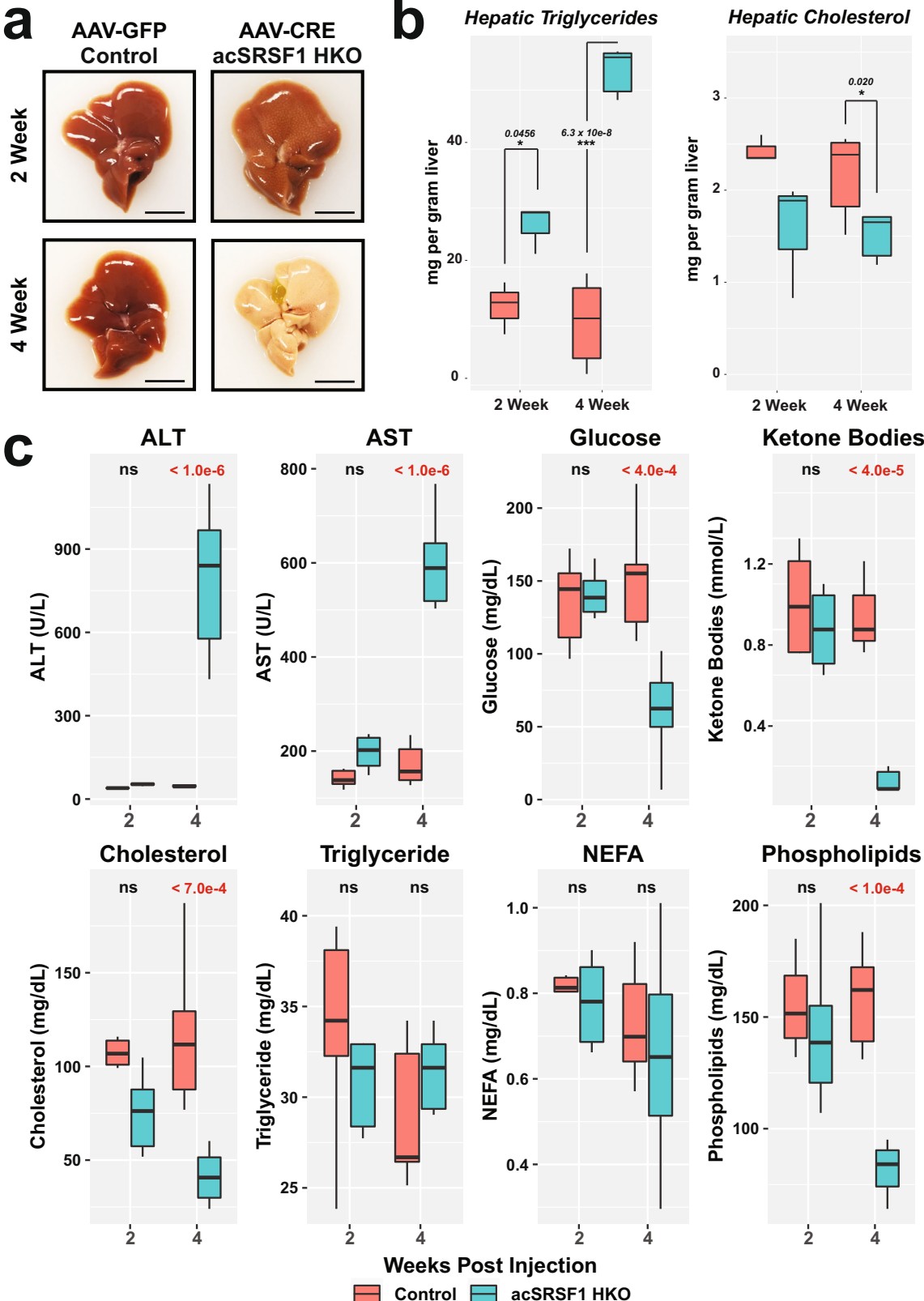

**Fig. 6 | Acute SRSF1 HKO mice develop hepatic steatosis and liver damage.**
**a** Representative gross images of whole livers harvested from control and acSRSF1 HKO mice at the indicated timepoints. Scale bar = 1 cm. **b** Box plot of measured hepatic triglyceride and cholesterol levels in control and acSRSF1 HKO mice (*n* = 3 biologically independent animals per group at 2 weeks; *n* = 6 and 5 for control and acSRSF1 HKO at 4 weeks, respectively). A two-tailed Student t-test was performed to test for significance. Significant differences are denoted by "*" with the p-value

listed above. **c** Profiling of the indicated metabolic parameter in serum collected from control and acSRSF1 HKO after 6 hours of fasting at the 2- and 4-week timepoints (*n* = 10 and 11 biologically independent animals for control and acSRSF1 HKO at 2 weeks, respectively; *n* = 8 biologically independent animals per group at 4-weeks). A two-way ANOVA statistical test was used to determine the significance between 2 groups at 2 timepoints. *P*-values are listed above each comparison. Source data are provided as a Source Data file.

lipids (Supplementary Fig. 4d). These findings explain the decreased serum phospholipid concentrations since they are mainly circulated within lipoprotein particles (Fig. 6c). Finally, no significant differences were observed in serum concentrations of triglycerides and non-esterified fatty acids. Altogether, these results suggest that hepatic loss of SRSF1 leads to severe metabolic insufficiency and hepatic damage.

## SRSF1 deficiency leads to global defects in the transcriptome

Given that SRSF1 regulates multiple aspects of mRNA metabolism, we reasoned that widespread defects to the transcriptome drive the cellular demise upon SRSF1 knockout. To effectively identify early transcriptome defects, we performed an RNA-seq study on hepatocytes isolated from acSRSF1 HKO at the 2-week time point. Unexpectedly, we found 3717 mRNA transcripts changing in abundance and 2996 transcripts changing in splicing following acute ablation of SRSF1 in adult hepatocytes (Supplementary Fig. 5a–c). A sample of skipped events was validated by RT-PCR (Supplementary Fig. 6a). The measurements by RT-PCR strongly correlate with the ΔPSI estimations determined by rMATS (Supplementary Fig. 6b). Compared to the SRSF1 HKO model, these changes correspond to a 104% and 252% increase in the number of mRNA abundance and splicing events, respectively. While we understand that comparisons between these two datasets are not ideal, nonetheless, we were surprised by the number of transcriptome alterations in the acSRSF1 HKO model. Of the mRNAs changing in abundance, ~68% were upregulated, corresponding to 2525 transcripts. In terms of alternatively spliced events, we detected changes in 2040 skipped exons, 252 retained introns, 354 mutually exclusive exons, 156 alternative 3′ splice sites, and 194 alternative 5′ splice sites. As expected, over two-thirds of skipped exons exhibited decreased inclusion since SRSF1 tends to promote exon inclusion. Overall, these findings provide strong evidence that the transcriptome defects caused by SRSF1 knockout precede the onset of damage and inflammation.

To further explore if the observed splicing defects in the acSRSF1 HKO model are directly regulated by SRSF1, we performed eCLIP-seq to profile SRSF1-binding distributions on transcripts in hepatocytes. Two independent eCLIP libraries were prepared and sequenced with excellent reproducibility between the replicates, indicated by a high correlation in gene read counts ($R_{spearman} = 0.941$, p-value $<1 \times 10^{-15}$). Using the CLIPper peak calling tool, we detected a total of 5,272 SRSF1 binding peaks between the two replicates[42]. We found nearly two-thirds (~64%) of the binding peaks mapped to either exon or exon-intron boundary sequences (Supplementary Fig. 5d). In addition, gene ontology analysis of SRSF1-bound transcripts showed enrichment for RNA binding proteins and metabolic processes such as lipoprotein particle and sterol transport (Supplementary Fig. 5e). Importantly, SRSF1 binding peaks in mouse hepatocytes were enriched with the GAAGAA consensus sequence motif established in previously published SRSF1 binding studies (Supplementary Fig. 5f). Next, we identified SRSF1 binding peaks that were in proximity to DSEs found in acSRSF1 HKO mice. If the misspliced exons in acSRSF1 HKO were dependent on SRSF1's splicing activity, there would be an enrichment of SRSF1 binding peaks in and around DSEs. To our surprise, less than 6% of the binding peaks localized within DSEs (Supplementary Fig. 5g). Conversely, examining the spatial binding distribution of SRSF1 on the 2040 differentially skipped exons in acSRSF1 HKO revealed a striking de-enrichment in binding in comparison to a control set of constitutive exons (Supplementary Fig. 5h). Collectively, these findings suggest that the majority of transcriptome defects arising after acute SRSF1 deletion are independent of its splicing activity.

## The Unfolded Protein Response is not activated upon SRSF1 knockout

Given the presence of aberrant transcripts in acSRSF1 HKO and SRSF1's role in promoting NMD, a possible mechanism of hepatocyte failure is an accumulation of unfolded proteins leading to ER stress and activation of the unfolded protein response, or UPR[43–45]. To explore this hypothesis, we began by examining expression changes in acSRSF1 HKO of genes known to be upregulated in the setting of UPR activation (Supplementary Fig. 7a). However, as is evident from the heatmap, there was a downregulation of UPR-responsive genes, suggesting a lack of UPR activation. We also assessed UPR activation by measuring protein levels of the transcription factor CHOP and by quantifying the splicing of *Xbp1* mRNA into its short isoform[46,47]. We found that both CHOP protein levels and *Xbp1* mRNA splicing were unchanged in acSRSF1 HKO hepatocytes (Supplementary Fig. 7b, c), further confirming that ER stress and UPR activation are not the direct triggers of hepatocyte death in acSRSF1 HKO livers.

## Loss of SRSF1 leads to R-loop accumulation and widespread DNA damage

To gain further insight into the biological processes affected by the transcriptome perturbations in acSRSF1 HKO hepatocytes, we performed an integrated gene ontology analysis using genes changing in expression and splicing. We constructed a GO network where each node corresponds to an enriched biological process. The size and color represent the number of genes within each node and the overall direction of differential expression, respectively (Fig. 7a). Furthermore, nodes with a blue outline signify ≥10% of the associated genes exhibited differential splicing. Similar to the chronic model, acSRSF1 HKO displayed a strong enrichment for immune processes indicating that loss of SRSF1 in hepatocytes triggers the necessary signals for the recruitment and activation of inflammatory cells before the onset of damage. We also noticed the simultaneous induction of genes associated with regeneration, angiogenesis, and proliferation-related functions. With cell death and declining hepatic function(s) in acSRSF1 HKO, this is likely a compensatory response to ensure timely repopulation of the liver tissue with healthy hepatocytes.

The GO network also revealed strong upregulation and impaired splicing of genes associated with apoptosis and DNA repair pathways (Fig. 7a). Accordingly, we hypothesized that overwhelming DNA damage might be the primary cause of hepatocyte death in SRSF1-deficient livers. A previous study identified SRSF1 as a crucial factor in maintaining genome stability, wherein SRSF1 depletion in DT40 cells caused widespread and unresolvable DNA damage due to the accumulation of genotoxic RNA-DNA hybrid structures known as R-loops[25]. While these structures are considered byproducts of transcription, their occurrence in most cells is rare and readily resolved by dedicated enzymes[48]. Growing evidence suggests that co-transcriptional splicing factors, such as SRSF1, facilitate the release of nascent RNA from the template DNA to counteract R-loop formation.

To determine if DNA damage is present in acSRSF1 HKO hepatocytes, we performed immunofluorescent staining on liver tissue sections from Control and acSRSF1 HKO mice for γH2A.X, a well-known marker of DNA damage[49]. In agreement with our hypothesis, we observed robust staining of γH2A.X foci only in the hepatocyte nuclei of acSRSF1 HKO mice (Fig. 7b). Furthermore, γH2A.X-positive nuclei were readily detected in SRSF1 HKOs at the 10-day and 1-month time points as well. We also confirmed that the γH2A.X-positive nuclei were present in SRSF1-deficient hepatocytes by performing a co-staining of γH2A.X and SRSF1 (Supplementary Fig. 8). Next, we set out to confirm if SRSF1-deficient hepatocytes display elevated levels of R-loops. We performed a dot blot assay using the S9.6 monoclonal antibody, which recognizes RNA-DNA hybrids, on DNA isolated from Control and acSRSF1 HKO hepatocytes (Fig. 7c). SRSF1-deficient hepatocytes showed strikingly elevated levels of RNA-DNA hybrids, most likely due to the increased formation of R-loops.

Accumulation of R-loops can severely impair transcriptional dynamics. These highly stable hybrid structures can interfere with

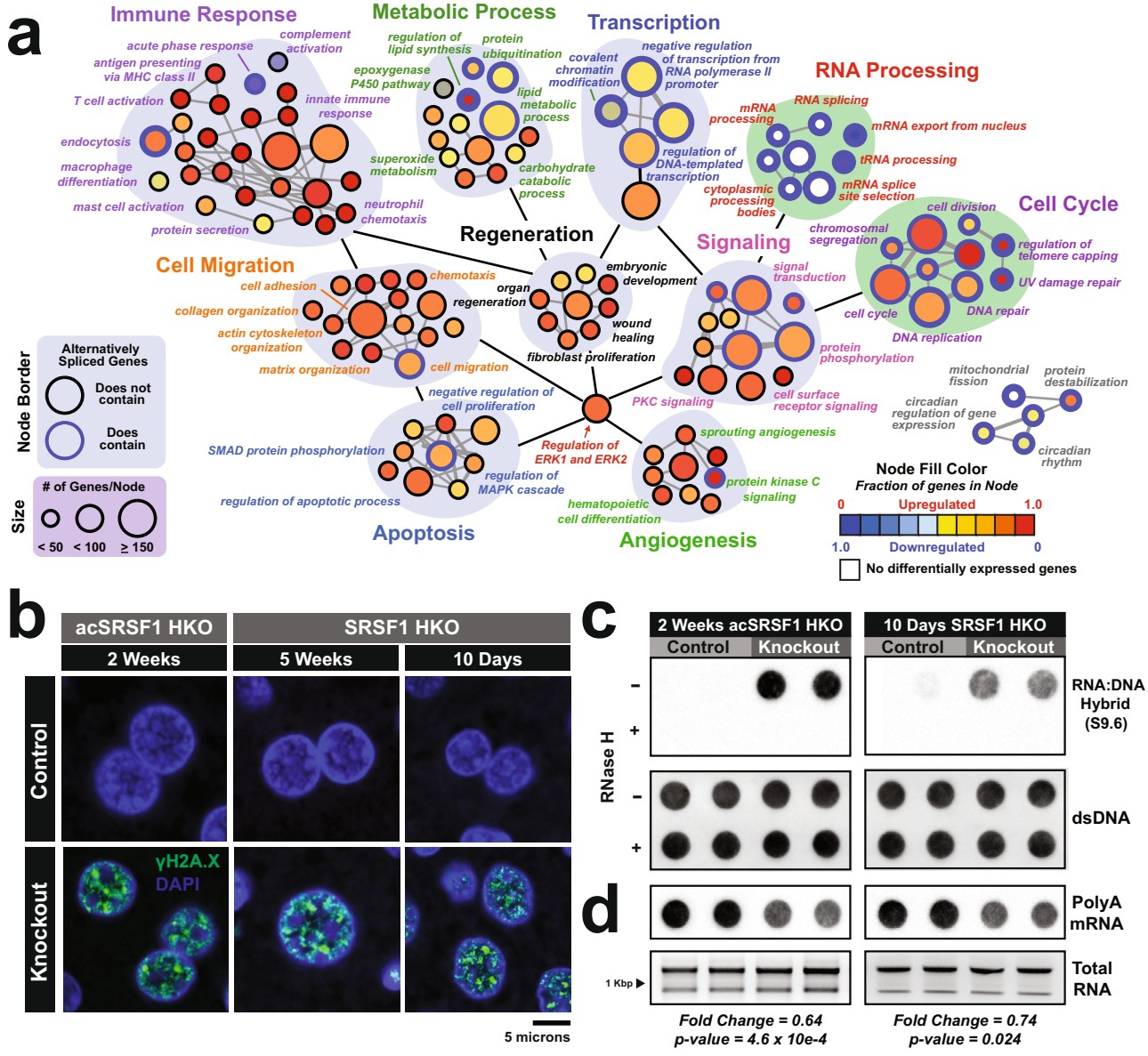

**Fig. 7 | Loss of SRSF1 results in DNA damage mediated by R-loop accumulation.**
**a** Gene ontology network map showing biological processes enriched for genes with differential expression or exon splicing in acSRSF1 HKO hepatocytes. Each node corresponds to an enriched GO term, with the size and color representing the number of genes within the node and the direction of differential expression, respectively. Nodes with blue outlines signify nodes that contain genes with differential splicing events in acSRSF1 HKO. **b** Representative IF images of liver sections from acSRSF1 HKO and SRSF1 HKO ($n = 5$ per group for 2- and 5-weeks; $n = 4$ per group for 10 days) at the indicated timepoints probed for γH2A.x (green), a DNA damage marker, with nuclear counterstaining using DAPI (blue).

**c** Representative dot blots showing the detection of R-loops in purified DNA from control and SRSF1 knockout samples ($n = 4$ per group) using the S9.6 antibody that recognizes RNA:DNA hybrids. As a negative control, half of each DNA preparation was treated with RNAse H. Corresponding dot blot for dsDNA was performed to confirm equal loading. **d** Dot blot analysis of polyA⁺ mRNA on purified total RNA using a polydT probe to assess relative steady-state levels of transcribed mRNA. Total RNA gel electrophoresis is shown for confirmation of equal loading. Values signify mean relative fold change between knockout and control. A two-tailed Student's t-test was performed to compare the differences in mean fold change. Source data are provided as a Source Data file.

transcription by directly blocking the activity of RNA polymerase. With R-loop-induced DNA damage in vivo, transcription is further impacted as DNA damage triggers phosphorylation and ubiquitination of RNA polymerase II, resulting in global transcriptional repression. To assess the global transcriptional activity of SRSF1-deficient hepatocytes, we performed a polydT dot blot assay on equally loaded total RNA and evaluated differences in steady-state levels of polyA mRNA between Controls and SRSF1-deficient hepatocytes (Fig. 7d). Since total RNA is primarily composed of rRNA (~80%), which has a longer half-life than mRNA (days versus minutes, respectively), we expected this assay would allow for a detectable decrease in polyA signal in the setting of global transcriptional repression. Indeed, a significant reduction in

polyA-to-total RNA signal ratio was observed in SRSF1-deficient hepatocytes relative to Control hepatocytes. Specifically, we detected a fold change in the signal ratio of 0.64 and 0.74 in 2-week acSRSF1 HKO and 10-day SRSF1 HKO mice, respectively. Collectively, these data provide compelling evidence that SRSF1 deficiency results in excessive accumulation of R-loops, resulting in subsequent DNA damage and global transcription repression.

## SRSF1-deficient hepatocytes display diminished global translation
Given the decreased level of total mRNA in SRSF1-deficient hepatocytes, we expected the hepatic proteome to be markedly affected.

To study changes in the hepatic proteome resulting from SRSF1 deficiency, we performed quantitative high-throughput mass spectrometry analysis on hepatocytes from control and acSRSF1 HKO mice. We detected a total of 3,603 distinct proteins, with 613 and 159 of the detected proteins significantly decreasing and increasing in relative abundance ($Log_2|IBAQ$ ratio| > 1, adjusted p-value <0.10), respectively (Supplementary Fig. 9a). To gain confidence in the differential protein abundances estimated from the proteomics data, western blot analysis was performed for RPS16, a protein randomly selected from the set of proteins changing significantly in abundance. The fold change measured by western blot for RPS16 agrees well with the proteomics data (Supplementary Fig. 6c). We also noted that aside from SRSF4, which was mildly elevated, all other SR protein levels remained stable and did not undergo a compensatory increase in abundance following SRSF1 deletion (Supplementary Fig. 9b). Overall, nearly 80% of differentially abundant proteins (DAPs) in acSRSF1 HKO hepatocytes exhibited reduced levels. Furthermore, it is known that SRSF1 can associate with

polyribosomes and promote the translation of target mRNAs in an mTOR-dependent manner[50,51]. However, we found that only ~16% of DAPs in acSRSF1 HKO demonstrated binding of SRSF1 on the associated transcript with no enrichment for downregulated proteins as would be expected (Supplementary Fig. 9c–e). Moreover, the intersection of DAPs with mRNAs changing in abundance and splicing showed a modest overlap of ~18% and ~14%, respectively (Supplementary Fig. 9f, g).

Next, we utilized gene ontology analysis to understand the various processes affected by the altered proteome of SRSF1-deficient hepatocytes (Supplementary Fig. 9c). Surprisingly, our results showed that proteins with decreased abundances strongly enriched for factors involved in translation. Specifically, a striking depletion was observed for ribosomal proteins, tRNA synthetases, and translation initiation factors with no significant change in expression of the associated mRNA (Fig. 8a). Because acSRSF1 HKO displayed widespread depletion of essential elements necessary for mRNA translation, we reasoned this

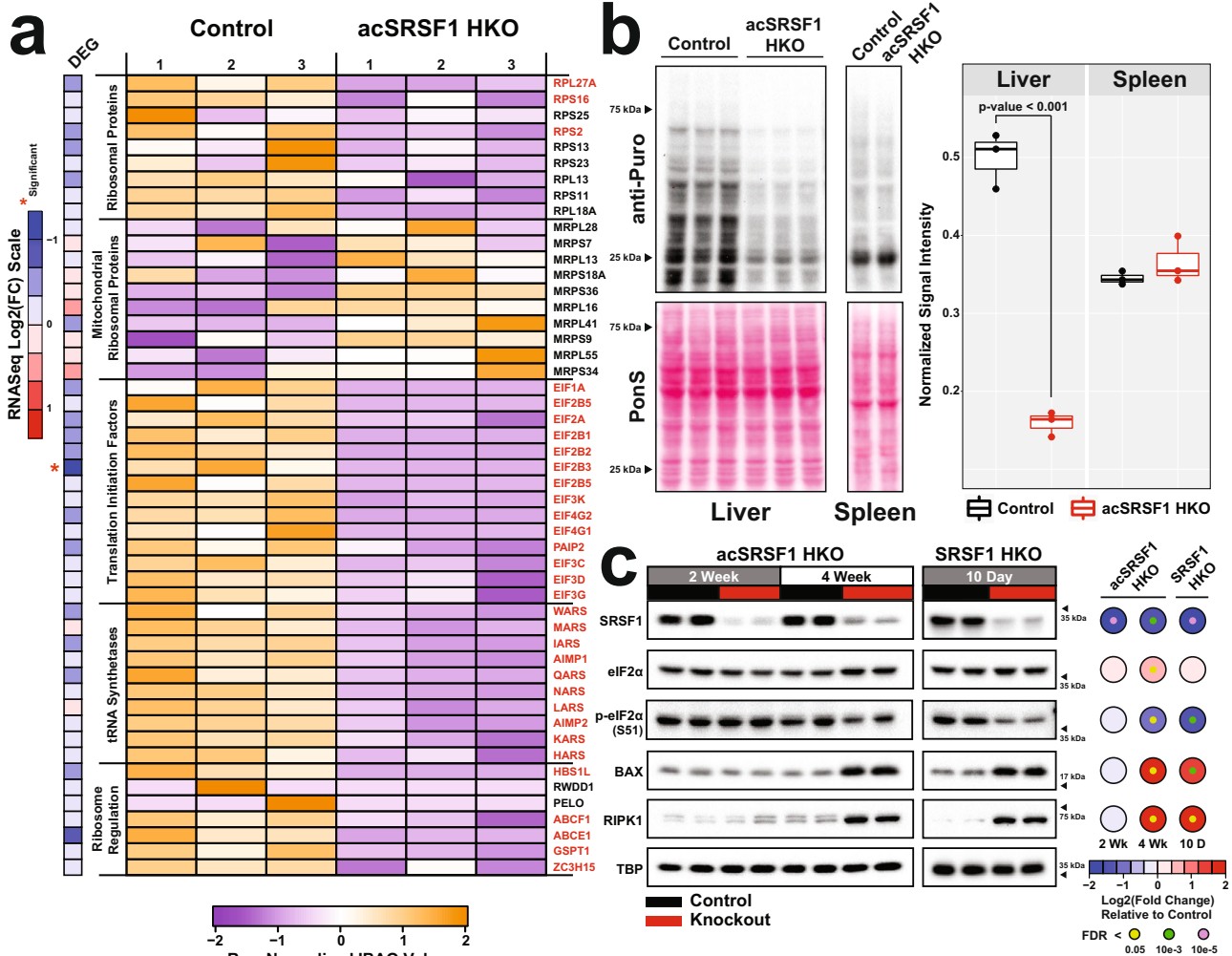

**Fig. 8 | Acute SRSF1 HKO hepatocytes display striking defects in global translation. a** Heatmap depicting relative protein abundances based on calculated IBAQ values (row normalized) of various factors involved in translation from mass spectrometry of control and acSRSF1 HKO hepatocytes ($n = 3$ biologically independent animals per group). The factor labeled in red indicates a significant difference in abundance ($Log_2|Fold Change| \geq 1$, FDR < 0.05) between control and knockouts. The heatmap strip on the left shows the relative fold change in mRNA abundance of the associated factor. **b** Measurement of de novo protein synthesis using a puromycin incorporation assay on hepatocytes isolated from control and acSRSF1 HKO at the 2-week timepoint ($n = 3$ biologically independent animals per

group). Spleens were harvested as well and were used as control tissue. Relative nascent protein synthesis was quantified as the total signal ratio of puromycin-labeled peptides (anti-puromycin) to total protein (Ponceau S). A two-tailed Welch two-sample t-test was performed. **c** Representative western blots of indicated factors on control and SRSF1 knockout hepatocytes isolated from 2- and 4-week acSRSF1 HKO and 10-day SRSF1 HKO mice models ($n = 4$ biologically independent animals per group). Heatmap displays quantification of the average fold change of protein abundance in SRSF1 HKO relative to controls. The colors of the large and small nodes signify average fold change and false-discovery rate, respectively. Source data are provided as a Source Data file.

would result in the failed assembly of the ribosomal complex. This was further validated by hepatocyte-specific polysome profiling, which showed an absence of polysomal peaks in acSRSF1 HKO, indicative of impaired ribosomal elongation (Supplementary Fig. 9h). To further quantify overall protein synthesis rates in vivo, we pulse-labeled control and acSRSF1 HKO mice at the 2-week time point with puromycin followed by immunoblotting with an anti-puromycin antibody (i.e., SUnSET assays)[52]. We detected a robust decrease in puromycin-labeled peptides in acSRSF1 HKO, signifying lower protein synthesis rates relative to control hepatocytes (Fig. 8b). Spleen was used as a control tissue which showed no difference in puromycin incorporation. The reduction in global protein synthesis could be attributed to a generalized stress response, which is facilitated by phosphorylation of the α subunit of the eukaryotic translation initiation factor 2 (eIF2α) at serine S51[53]. However, acSRSF1 HKO mice did not exhibit an increase in eIF2α phosphorylation (Fig. 8c). Instead, a decrease in phosphorylated eIF2α was noted in 4-week acSRSF1 HKO and 10-day SRSF1 HKO hepatocytes, which likely reflects a regeneration-associated surge in protein synthesis[54,55].

Because SRSF1 ablation resulted in severe depletion of mRNAs and proteins, we anticipated that SRSF1-deficient hepatocytes would likely become inactive and be eventually eliminated. Given the absence of apoptosis indicated by the negative TUNEL staining, we posited that SRSF1-deficient hepatocytes may be undergoing a necrotic death. This notion was supported by the histological features of swollen hypochromic cells with intact nuclei in SRSF1 HKO livers. To further confirm the mode of cell death in acSRSF1 HKO, we measured the levels of BAX and RIPK1 proteins, which are markers of apoptosis and necrosis, respectively (Fig. 8c)[56,57]. Interestingly, both BAX and RIPK1 were elevated in 4-week acSRSF1 HKO and 10-day SRSF1 HKO hepatocytes. These results indicate that SRSF1-deficient hepatocytes undergo necroptosis—a form of immunogenic programmed cell death—resulting in the release of damage-associated molecular patterns (DAMPs), which signals the recruitment of inflammatory cells and subsequent clearance of cellular debris[58]. Indeed, the features of inflammation were readily detected both histologically and in the transcriptomic signatures of SRSF1 HKO livers.

### Knockdown of SRSF1 in HepG2 reproduce acSRSF1 HKO pathology

We next tested if SRSF1 HKO pathology could be reproduced in the context of human cells. We examined publicly available transcriptome datasets from the ENCODE project of shRNA-mediated knockdowns of SR proteins in HepG2 cells, a human liver cancer cell line[59]. The differential gene expression and splicing analysis for SRSF1, SRSF3, SRSF5, SRSF7, and SRSF9 depleted cells showed that among all SR proteins, knockdown of SRSF1 had the greatest impact on the transcriptome (Supplementary Table 4). Gene ontology analysis revealed that downregulated genes in SRSF1-depleted cells were strongly enriched for factors involved in translation and the ribosome complex (Supplementary Fig. 10a, b). This is consistent with our findings of reduced ribosomal proteins and diminished global translation in acSRSF1 HKO hepatocytes.

We also noted that SRSF1 knockdown triggers the induction of many known p53-responsive genes (Supplementary Fig. 10c). In a healthy cell, p53 is maintained at low levels and remains inactive in a complex with MDM2[60]. However, upon DNA damage or other stresses, various pathways activate p53 by phosphorylation of its N-terminal domain. This phosphorylation facilitates the disassociation of p53 from MDM2, resulting in increased stability and activation of its transcriptional activity, thus, promoting the expression of DNA damage response genes[61]. Activation of p53 in the setting of SRSF1 depletion is consistent with the widespread DNA damage observed in SRSF1 HKO hepatocytes. As anticipated, both total and phosphorylated p53 levels in 10-day SRSF1 HKO were elevated relative to control hepatocytes (Supplementary Fig. 10d).

Next, we performed siRNA-mediated depletion of SRSF1 in HepG2 cells and found robust activation of p53 within 36 hours of SRSF1 knockdown (Supplementary Fig. 10e). Importantly, similar to SRSF1 HKO models, transient depletion of SRSF1 in HepG2 cells was sufficient to cause R-loop accumulation (Supplementary Fig. 10f), DNA damage, diminished protein synthesis, and eventual cell death (Fig. 9a). Finally, to test if p53 activation mediates the arrest in protein synthesis in SRSF1 depleted cells, we performed SRSF1 and p53 double knockdowns (Supplementary Fig. 10e). However, reducing p53 in SRSF1 depleted cells did not rescue the defects in protein synthesis or cell death, indicating these phenotypes are p53-independent (Fig. 9a; Supplementary Fig. 10f). The typical response of cells that accumulate DNA damage is to undergo cell cycle arrest to allow the cell to repair the damage. To examine if SRSF1-deficient cells also exhibit cell cycle arrest, we performed a propidium iodide (PI) flow cytometry analysis (Fig. 9b). Indeed, we saw that SRSF1-depleted cells arrest during the G2/M phase, which is indicative of damages accumulating during DNA replication that are beyond repair. Taken together, these results illustrate that the accumulation of detrimental R loops and subsequent DNA damage responses observed in the SRSF1 HKO mice models are recapitulated in SRSF1-depleted human cells.

## Discussion

Since its initial discovery as a splicing factor, SRSF1 has been recognized for its myriad roles in mRNA metabolism, including stability, export, NMD, and translation[24,62]. While SRSF1 is well-characterized biochemically, exactly how its activities impact in vivo tissue function(s) is still not fully appreciated. In this study, we define a crucial role for SRSF1 in maintaining genome integrity, and we demonstrate how the accumulation of R-loops in its absence affects hepatocellular gene expression, triggers severe metabolic dysfunctions, and leads to NASH-like liver pathology. Although we initially set out to understand the role of SRSF1 in maintaining liver physiology, we discovered its broader core function in maintaining cell viability.

### Caveats of liver-specific transgenic mice

We began our investigations by studying the effects of SRSF1 ablation on hepatocyte function. Because the whole-body knockout of SRSF1 is embryonically lethal, we generated a Cre-mediated hepatocyte-specific knockout referred to as SRSF1 HKO[26]. We were surprised to find that SRSF1 HKO mice survived and could maintain viable, functional livers, despite early insufficiencies and damage. However, on further examination, we determined that SRSF1-deficient hepatocytes were undergoing rapid cell death, triggering a compensatory regenerative response. Even more astonishing was that SRSF1 HKO mice livers ultimately evaded the deletion and repopulated their parenchyma with SRSF1-expressing hepatocytes. This repopulation reversed the liver damage, inflammation, and initial growth delays seen in SRSF1 HKO mice over time.

We found that this resistance to knockout in SRSF1 HKO was due to the suppression of Cre expression[39]. As for the mechanistic underpinnings of this phenomenon, the work by Duncan et al. (2012) provides the most plausible explanation[63]. A unique feature of liver tissue is that it comprises polyploid or aneuploid hepatocytes, thus, creating increased genetic diversity[64,65]. Duncan et al. elegantly showed that certain conditions of chronic liver injury result in the selection of a differentially resistant aneuploid karyotype leading to adaptation[63]. With SRSF1 HKO mice, knockout-resistant aneuploid hepatocytes—that lack the Cre transgene—are likely repopulating the liver. Such an effect is possible in the liver, which has a high regenerative capacity and a number of aneuploid cells. On the contrary, ablation of SRSF1 in cardiomyocytes, which have low regenerative potential, resulted in heart failure and death[26]. Given that the whole-body deletion of Srsf3 is embryonically lethal, it is highly likely that SRSF3 HKO mice would also exhibit the adaptive mechanisms seen in SRSF1 HKO. In fact,

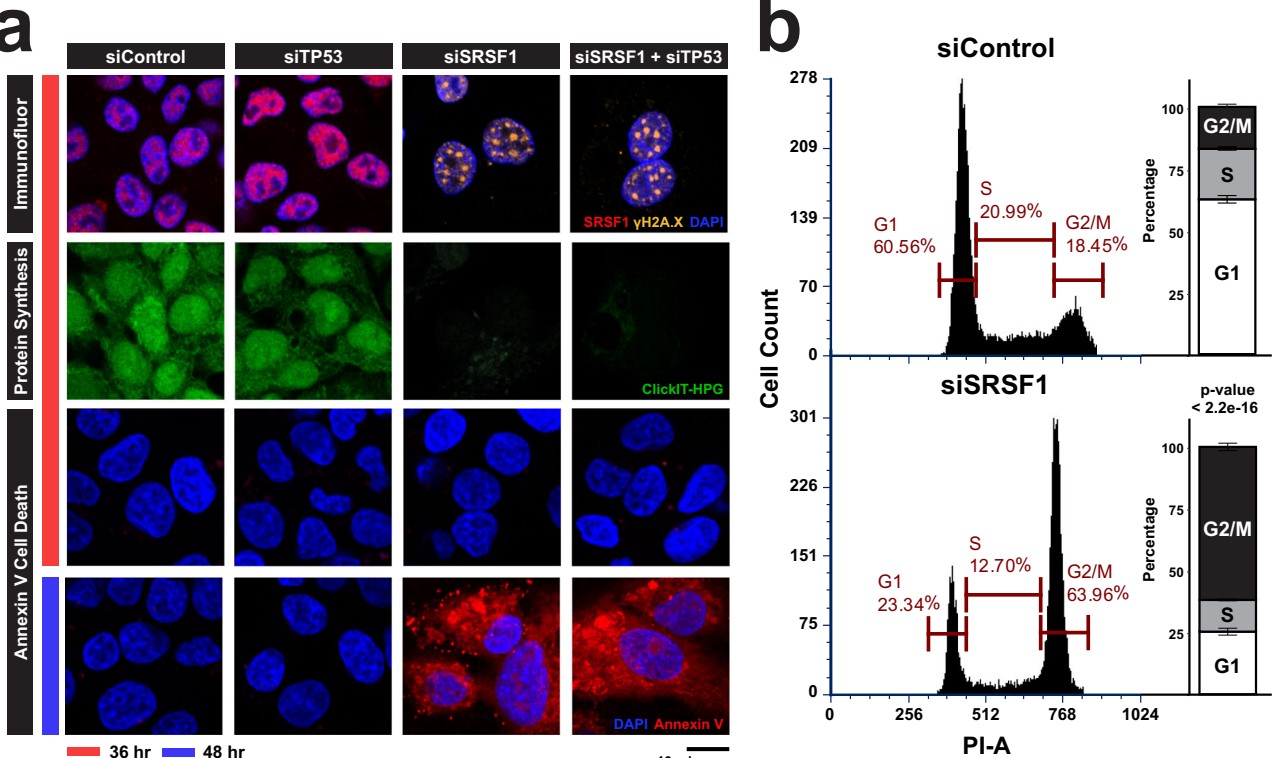

**Fig. 9 | Acute knockdown of SRSF1 in HepG2 recapitulates SRSF1 HKO pathology. a** HepG2 cells were cultured and treated with the indicated siRNA for 36 and 48 h before performing the specified assays. First Row: Immunofluorescent staining for SRSF1 (red), γH2A.X (yellow), and nuclear (blue) after 36 h of knockdown with the specified siRNA. Second Row: Qualitative assessment of nascent protein synthesis (green) of the same field shown in the first row using a fluorescent-based ClickIT-HPG incorporation assay. Third and Fourth Rows: Annexin V staining (red) for cell death of HepG2 cells at 36 and 48 h post

knockdown by the specified siRNAs. All images shown are representative of $n = 3$ replicates biologically independent cell culture experiments, 5 fields per experiment. **b** Representative distribution of cell cycle phase determined by PI flow in HepG2 cells treated with the indicated siRNA for 72 h ($n = 3$ biologically independent cell culture experiments). Quantification is shown on the right with values displayed as mean ± SEM. A Chi-squared test for proportions was performed between the control and experimental group. Source data are provided as a Source Data file.

the knockdown of SRSF3 in a human HCC cell line robustly down-regulates SRSF1 protein levels due to its aberrant splicing[66]. Thus, our work illustrates a potential limitation to consider when using liver-specific transgenic animal models.

**Mechanism of demise in SRSF1 depleted hepatocytes**

We performed a series of experiments to understand the cascade of molecular events leading to hepatocyte death in SRSF1 HKO. Elucidating the mechanism was complicated by the fact that SRSF1 regulates multiple aspects of gene expression. To help dissect the pathogenic process, we developed an acute hepatocyte-specific SRSF1 knockout mouse model, which allowed for precise temporal control of SRSF1 deletion. As expected, acSRSF1 HKO mice displayed robust hepatocyte cell death with subsequent development of hepatic damage. Importantly, this model permitted us to probe primary changes in SRSF1-deficient hepatocytes before the onset of damage. Using this model, we discovered that SRSF1 depletion caused widespread DNA damage (Fig. 10). An often overlooked attribute of SRSF1 is its essential function in maintaining genomic stability. It protects the genome by preventing the formation and accumulation of deleterious R-loops that naturally occur during transcription. In fact, this activity was discovered when depletion of SRSF1 using a tet-repression system in DT40 chicken B cells resulted in extensive cell death followed by an expansion of SRSF1-expressing tet-resistant colonies[25]. The DT40 cell culture model strongly parallels our findings in SRSF1 HKO mice. Particularly, we found that SRSF1-deficient hepatocytes exhibit reduced transcriptional activity, likely due to the combined inhibitory effects of

R-loop accumulation and DNA damage response. This ultimately results in diminished protein synthesis, causing depletion of essential enzymes and hepatic proteins, which impairs the metabolism and transport of lipids. We further demonstrated that lipid-laden, metabolically challenged, SRSF1-deficient hepatocytes succumb to necroptotic cell death, triggering inflammation, and fibrosis, thereby provoking a NASH-like liver pathology (Fig. 10). It would be interesting to explore in the future whether under certain circumstances SRSF1 expression/activity is diminished in human hepatocytes and if that is clinically relevant in the development of fatty liver disease.

In conclusion, our work underscores a fundamental role for SRSF1 in maintaining genome integrity and cell viability. While our study investigates the effects of SRSF1 depletion in hepatocytes, our findings are broadly applicable to any eukaryotic system. Several groups have recently examined the role of SRSF1 in T cell development, vascular smooth muscle proliferation, and skeletal muscle development utilizing cell-type-specific knockout mice models[27,29–31]. Although the observed outcomes of SRSF1 ablation in these studies were attributed to its alternative splicing activity, our findings suggest these effects are much more complex and potentially secondary to genome instability. For instance, reduced mRNA translation in SRSF1-deficient cells would inhibit NMD and deplete spliceosomal factors that can indirectly lead to missplicing and/or accumulation of aberrantly spliced transcripts. Also, while it is possible that missplicing of some crucial transcripts exacerbates the phenotypes of SRSF1-depleted cells, it is unlikely the primary and the only contributing factor. Our data instead indicates that the DNA damage

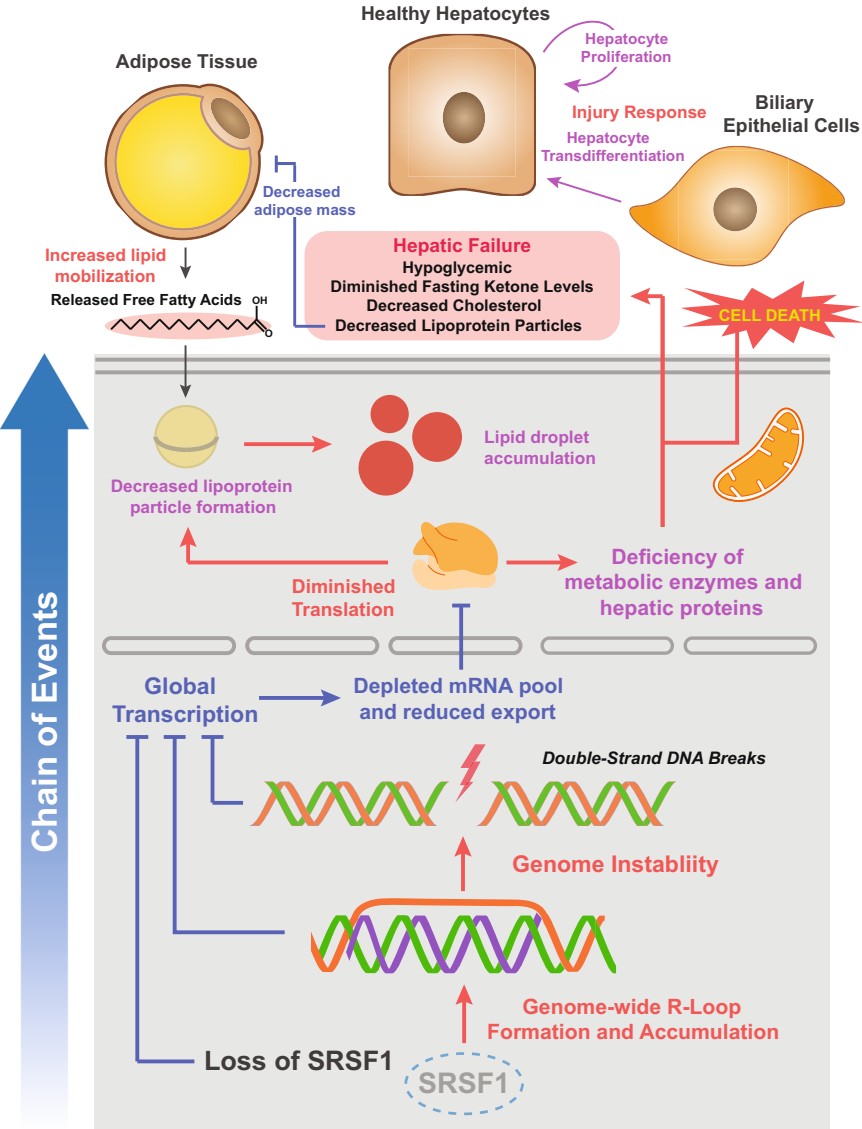

**Fig. 10 | Proposed mechanism of cell death in SRSF1-deficient hepatocytes.**
Loss of SRSF1 initially leads to an overwhelming accumulation of R-loops within the actively transcribed DNA of hepatocytes. This results in widespread DNA damage as unresolved R-loops are prone to double-stranded breaks. The combined effects of SRSF1 activity loss, R-loop accumulation, and DNA damage response causes a halt in global transcription. With decreased mRNA synthesis and export, there is diminished translation and protein production which leads to subsequent cell death and hepatic failure.

caused by the accumulation of R-loops is the main driving force for cellular damage, resulting in many downstream secondary effects. Indeed, there is growing evidence that RNA processing factors serve key roles in protecting the genome from the deleterious effects of transcription[67]. Initial studies in yeast revealed that disruption of mRNA export factors results in R-loop-mediated DNA damage[68]. This effect has since been shown to result from the disruption of multiple factors involved at various stages of RNA biogenesis[69,70]. Recently, it was reported that the knockdown of Slu7, another splicing regulator, in cultured cells and mice livers causes R-loop accumulation and DNA damage[66]. Intriguingly, depletion of Slu7 also evoked missplicing of SRSF1 transcript and subsequent downregulation of SRSF1 protein. Thus, our work supports the emerging notion of RNA processing factors moonlighting as guardians of the genome.

## Methods

### Generation of SRSF1 HKO and acSRSF1 HKO

*Srsf1*flox/flox mice were crossed with transgenic *AlbCre* mice (C57BL6/J background; The Jackson Laboratory, Bar Harbor, Maine, USA), in which expression of Cre recombinase is driven by the hepatocyte-specific albumin (*ALB*) promoter[26,37]. The resulting progeny (*Srsf1*flox/+ *AlbCre*+/−) were mated with *Srsf1*flox/flox mice to generate *Srsf1*flox/flox; *AlbCre*+/−, or hepatocyte-specific SRSF1 knockout mice (SRSF1 HKO). To generate control mice (*AlbCre*+/−), *AlbCre*+/+ were mated with wild-type C57BL/6 J mice. Genotyping was performed on genomic DNA isolated from tail clipping using primers and protocols described by The Jackson Laboratory. For the acute hepatocyte-specific deletion of SRSF1 (acSRSF1 HKO) mice model, *Srsf1*flox/flox mice were injected with adeno-associated viral vectors (VectorBio Labs) either expressing GFP (Controls) or the Cre recombinase (acSRSF1 HKO) driven by the hepatocyte-specific thyroxine-binding globulin (TBG) promoter. Mice were injected at 8 weeks of age with a viral titer of $5 \times 10^{11}$ genome copies (gc) via the tail vein. For the acSRSF1 HKO mice, timepoints are in reference to the time elapsed after viral transduction. Mice were housed on a standard 12-hour-light/dark cycle (18-23 °C ambient temperature; 40-60% humidity) and were allowed *ad libitum* access to water and a normal chow diet (2918 Envigo Teklad). The mice were fasted for 6 hours prior to harvesting the tissues or blood. National

Institutes of Health (NIH) guidelines for the use and care of laboratory animals were followed, and all experiments were approved by the Institutional Animal Care and Use Committee at the University of Illinois at Urbana-Champaign (Champaign, IL).

## Blood collection and serum chemistry assays

Mice were fasted about 10 hours prior to blood collection. Blood was collected from the retro-orbital venous sinus using EDTA-coated blood-collecting capillaries. Mice were temporarily anesthetized using isoflurane before collecting blood. For nonterminal procedures about 200 μL was collected otherwise about 600 μL was collected. Blood from the capillary was transferred into BD microtainer tubes and then centrifuged according to the manufacturer's protocol to separate the serum. Serum was then transferred into 1.5 mL microcentrifuge tubes and snap-frozen in liquid nitrogen before storing at −80 °C. Serum cholesterol, triglycerides, ALT, and AST activity were measured using colorimetric assay kits provided by Infinity (Thermo Scientific). Serum bilirubin, both direct and indirect, was measured using the Sigma Bilirubin Assay kit (MAK126 Sigma) according to the manufacturer's protocol. Snap-frozen serum (100 μL) collected from control and acSRSF1 HKO mice were submitted to the Mouse Metabolic Phenotyping Core at the University of Cincinnati for serum analysis. A serum chemistry panel was obtained for each sample with measured concentrations of serum triglycerides, cholesterol, phospholipids, non-esterified fatty acids, glucose, ketones, ALT, and AST.

## Histology, immunohistochemistry, and immunofluorescence staining

**Tissue processing and embedding.** For paraffin embedding, harvested liver tissue pieces were washed in 1X phosphate buffered saline (PBS) and then immediately fixed in 10% neutral-buffered formalin (NBF) overnight at 4 °C. Fixed tissues were processed through a series of solvents starting from ethanol solutions to xylenes and then embedded into paraffin blocks using standard protocols. For cryo-embedding, washed tissue pieces were positioned into molds containing OCT medium (Tissue-Tek, Sakura) and then frozen using liquid nitrogen.

**H&E and Sirius Red staining.** Paraffin-embedded tissues sections (5 μm) were cut, placed onto microscope slides, and incubated in a series of xylene and alcohol dilutions to deparaffinize and rehydrate the sections in preparation for further staining. For hematoxylin and eosin (H&E) staining, standard procedures were followed using Modified Harris Hematoxylin Solution (7211 Richard-Allan Scientific) and Eosin Y Solution (7111 Richard-Allan Scientific). For Sirius red staining, sections were incubated in Picro-Sirius red solution (0.1% w/v Direct Red 80, 365548 Sigma-Aldrich, in a saturated aqueous solution of picric acid) for 1 hour. They were then briefly rinsed in two changes of 0.5% acetic acid solution. Stained sections were mounted with a coverslip using Permount (SP15-100 Fisher Scientific) mounting media.

**Oil red O staining.** OCT-embedded tissues were sectioned (10 μm) using a cryostat, placed onto a microscope slide, and then fixed in NBF for 15 minutes. Slides were brought to 60% isopropanol solution and then stained with freshly prepared Oil Red O solution (0.3% w/v Oil Red O in 60% isopropanol) for 15 minutes. Once stained, the slides were rinsed in 60% isopropanol solution and gently counterstained with Modified Harris Hematoxylin. Slides were rinsed with distilled water and mounted using CC mount (C9368 Sigma) and coverslip.

**Immunohistochemistry (IHC) and immunofluorescent (IF) staining.** Paraffin sections were deparaffinized, rehydrated, and then antigen retrieved in Tris-EDTA buffer (10 mM Tris, 1 mM EDTA, 0.05% Tween 20, pH 8.0) using a slow cooker at 120 °C for 10 min. The sections were then incubated in wash buffer containing 1X tris-buffered saline (TBS) and 0.025% Triton X-100 and then blocked using 1X TBS, 10% Normal Goat Serum (NGS), and 1% BSA for 2 hours at room temperature (RT). Primary antibodies were applied to the sections at standardized concentrations and incubated overnight at 4 °C. Next, the sections were washed with wash buffer, and secondary HRP-conjugated or fluorescent antibodies were applied for 1 hour at RT for IHC or IF, respectively. For IHC, sections were washed in 1X TBS and then developed using a DAB Peroxidase Substrate kit (Vector Laboratories) for approximately 5 minutes. Sections were counterstained with hematoxylin, dehydrated into xylenes, and then mounted using Permount. For IF, sections were washed in 1X TBS, stained for the nucleus using ToPro3 (R37170 Thermo Fisher Scientific) for 15 min at RT, and then mounted using CC aqueous mounting media. All antibodies used and respective dilutions are listed in Supplementary Table S5.

**Imaging and analysis.** Histology and IHC slides were imaged on a Hamamatsu Nanozoomer, and IF slides were imaged using a Zeiss LSM 710 microscope at the Institute of Genomic Biology (IGB) core facility, UIUC. Counting of stained nuclei was performed using built-in thresholding and watershed functions available in Fiji, an open-source image processing program.

## Scoring of histological liver sections

All histologically stained sections were examined and scored using a two-headed microscope. The NAFLD Activity Scoring (NAS) index, along with a separate score for fibrosis, was obtained for each specimen based on published methods[71]. Slides were scored independently in a blinded fashion by two pathologists. Both investigators had to agree for a score to be given.

## Isolation and purification of hepatocytes

Hepatocytes were isolated and purified using the two-step collagenase perfusion technique[72]. Mice were first anesthetized with isoflurane and then secured to a surgery pad with the ventral side up. A "U" shaped incision was made on the abdomen to expose the liver. The liver was then perfused through the portal vein with 50 ml of wash buffer containing 1X Hanks Balanced Salt Solution (HBSS) and 1 mM EDTA (pH 8.0), without calcium and magnesium salts. Following this, the livers were perfused using 50 ml of digestion buffer containing 1X HBSS, 0.5 mM CaCl$_2$, 40 μg/mL soybean trypsin inhibitor, and 60 U/mL of Collagenase Type I from Worthington. The perfused liver was carefully excised out from the abdomen and transferred into a petri dish containing 1X HBSS. Using cell scrapers, the tissue was carefully massaged to release the cells from the capsule. The crude cell prep was then filtered through a 40 μm mesh filter and the resulting single cell suspension was centrifuged at 50 g for 5 minutes at 4 °C. The supernatant containing non-parenchymal cells and dead hepatocytes was discarded while the pellet containing live hepatocytes were resuspended in fresh 1X HBSS. The centrifugation wash was repeated two additional times before being aliquoted into 1.5 mL microcentrifuge tubes, flash frozen in liquid nitrogen, and stored in −80 °C until further use.

## RNA-seq library preparation, sequencing, and analysis

Total RNA was prepared from frozen hepatocyte pellets using the RNeasy tissue mini-kit (Qiagen). Downstream RNA quality was assessed using an Agilent Bioanalyzer and quantified using a Qubit Fluorometer by the Functional Genomic Core at the Roy J. Carver Biotechnology Center, UIUC. Hi-Seq libraries were prepared and 100-bp paired-end Illumina sequencing was performed on a HiSeq 4000 at the High Throughput Sequencing and Genotyping Unit, UIUC. RNA-Seq reads were processed for quality and read length filters using Trimmomatic (version 0.38)[73]. For differential gene expression analysis, RNA-Seq reads were psuedoaligned using Kallisto (version 0.44.0)[74]. Transcript abundances were converted to gene abundances using tximport (version 1.11.7)[75]. With gene abundance tables, differential gene

expression analysis was performed using DESeq2 (version 1.23.10)[76]. For differential splicing analysis, RNA-Seq reads were trimmed to a set length and then mapped using STAR (version 2.4.2a) onto the mouse vM19 genome (mm10) available from Gencode (https://www.gencodegenes.org)[77]. Alignment files were then used to perform differential splicing analysis using rMATS (version 3.2.5) and significant events were identified using imposed cutoffs (FDR < 0.10, junction read counts ≥ 10, and deltaPSI ≥ 15%)[78]. Gene ontology analysis was performed using gProfiler and Enrichr[79,80]. Filtering and processing of data was performed using custom Python and R Scripts. Details regarding RNA-seq sample information are provided in Supplementary Table 8.

### Protein isolation and western blot analysis

Total proteins were isolated from ~50 mg of snap-frozen liver tissue or purified cell pellet by homogenizing in 400 μL of cold homogenization buffer containing 10 mM HEPES (pH 7.5), 0.32 M Sucrose, 5 mM EDTA, 1% SDS, 5 μM MG132 and Pierce Protease Inhibitors (1 tablet per 10 mL of buffer, Catalog # A32953). Samples were sonicated in a water bath to shear DNA and clarified by centrifugation. Protein concentration was determined using the Pierce™ BCA Protein Assay Kit (Thermo Scientific). Protein lysates were diluted to 5 mg/mL and then boiled in 1X Laemmli buffer at 100 °C for 10 minutes. For cultured cells, samples were directly lysed in 2X Laemmli buffer (200 μL/well for 6-well plate), sonicated, and then boiled. After boiling, samples were cooled to room temperature and ~50 μg of proteins were resolved on a 10% SDS-PAGE gel and transferred using a wet transfer setup onto a PVDF membrane with 0.45 μm pore size (Immobilon, Millipore). Membranes were blocked using Tris-buffered saline containing 5% nonfat dry milk and 0.1% Tween 20 (TBST). After blocking, membranes were incubated with primary antibody overnight at 4 °C. The membranes were then washed with TBST to remove any unbound primary antibody followed by incubation with an appropriate horseradish peroxidase-conjugated secondary antibody for two hours. Membranes went through a final TBST wash and then visualized on the ChemiDoc XRS + using the Clarity Western ECL kit (BioRad). Densitometric quantification of the western blots were performed using Image Lab (version 6.1).

### Hepatic lipid isolation and quantification

Lipids were extracted from approximately 100 mg of liver tissue using the Folch Method[81]. Briefly, the tissue was homogenized in 1 mL of 2:1 choroform:methanol mixture. The homogenized mixture was incubated overnight at room temperature to allow for complete extraction. The mixture was then centrifuged at 10,000× g for 10 minutes to pellet any debris. The supernatant was transferred to a fresh tube. The interface was washed with 1 X PBS solution to remove any additional salt. The mixture was centrifuged at 4000× g, and the aqueous layer was removed. The organic phase was dried under a nitrogen stream and then reconstituted in 300 μL of ethanol. The reconstituted mixture was used with Infinity Kit to determine concentrations of triglycerides and cholesterol.

### Global proteomics analysis by mass spectrometry

Samples were processed for proteomics according to the FASP protocol and desalted on in-house prepared C18 tips[82]. The peptides were resuspended to 0.5 μg/μL, and 1 μg was separated on an EASY Spray C18 column (50 cm × 75 cm, 2 μm particle size) (ThermoFisher Scientific, Toronto) using an EASY nLC-1200. The mobile phase was composed of 0.1% formic acid in water (A) and 80% acetonitrile with 0.1% formic acid (B). The gradient was as follows: 5−40% B (0−120 min), 40%−100% B (120−125 min), 100% B (125−135 min). The peptides were analyzed on a Thermo QExactive HF mass spectrometer in a Top 20 data-dependent acquisition mode. Proteins were identified using MaxQuant to search the mouse (UP000000589) proteomes from UniProtKB (February 2019). Peptide spectral matches and protein False Discovery Rates were set to 1% and required a minimum of 1 unique peptide for identification. To increase the number of identified matches, a match between runs was enabled with a match time window of 0.7 minutes. Protein abundances were calculated using the iBAQ algorithm in MaxQuant. Differential protein abundance was performed using student's t-test with Bonferroni correction on iBAQ values between control and acSRSF1 HKO.

### Global translation quantification by SUnSET assay

Translating proteins were labeled using a protocol adapted from the Surface Sensing of Translation (SUnSET) method[52]. Mice were injected with puromycin prepared in sterile PBS with a dosage of 0.04 μmol per gram body weight. After 45 minutes, livers were harvested, and protein lysates were prepared as described previously. Proteins were separated by 10% SDS-PAGE and then transferred onto a PVDF membrane. Puromycin-labeled peptides were identified using the mouse monoclonal antibody 12D10 (EMD Millipore Catalog# MABE343). Relative protein synthesis levels were determined by densitometry analysis of whole lanes.

### Dot blot assays for DNA-RNA hybrid and polyA mRNA

DNA-RNA hybrid dot blot assay was adapted from a previously published report[83]. Briefly, DNA was isolated from approximately 50 mg of liver tissue or snap-frozen cells using DNeasy Blood & Tissue Kit (Qiagen) using the manufacturer's protocol. For each sample, 250 ng of DNA was digested with 5 Units of RNase H (NEB) as control. A slot blot apparatus was used to dot blot 250 ng of DNA onto a Hybond™-N + membrane (Amersham). The membrane was auto-cross-linked with UV (254 nm, 1200 mJ/cm²). It was then air-dried before beginning blocking with 5% nonfat dry milk and 0.1% Tween 20 (TBST) at RT for 30 minutes. The membrane was then incubated overnight at 4 °C with S9.6 monoclonal antibody diluted in blocking buffer. Finally, it was probed with HRP-conjugated anti-mouse secondary antibody, washed with TBST, and visualized on the ChemiDoc XRS + using the Clarity Western ECL kit (BioRad). All antibodies used and respective dilutions are listed in Supplementary Table 5.

Dot blot analysis for polyA mRNA was performed on total RNA to indirectly assess the relative activity of global transcription. Total RNA was isolated from approximately 50 mg of snap-frozen liver tissue using the RNeasy kit following standard protocols. 250 ng of total RNA was blotted onto the membrane as described previously for the DNA-RNA hybrid assay. The membrane was UV-crosslinked and air-dried before continuing. The blot was prehybridized with ULTRAhyb-Oligo Hybridization Buffer (Thermo Fisher Scientific) for 10 minutes at 42 °C with gentle agitation in a hybridization oven. PolyA mRNA was detected using biotinylated oligo-dT probes (Promega #Z5261). The probe was diluted in ULTRAhyb-Oligo Hybridization Buffer (0.5 μL of 5 pmol/uL per mL of buffer). Blot was incubated with a probe solution at 42 °C for 1 h. Blot was then washed twice with ULTRAhyb-Oligo Hybridization Buffer. Chemiluminescent detection was performed using the Chemiluminescent Nucleic Acid Detection Module Kit (Thermo Fisher Scientific # 89880) following the manufacturer's protocol.

### Cell culture and siRNA knockdown experiments

HepG2 cell line was obtained from ATCC (catalog HB-8065) and cultured according to ATCC specifications. Cells were cultured in DMEM supplemented with 10% FBS, 2 mM glutamine, and 10 U/ml penicillin and streptomycin. For knockdown experiments silencer select siRNAs against *SRSF1 and TP53* (Thermo Fisher Scientific #4392420 and #4390824) along with a negative control (Thermo Fisher Scientific #4390843) were used. Approximately 500,00 cells were seeded into a 6-well format and were reverse transfected with 20 nM of gene-specific siRNA oligos using RNAiMax (Thermo Fisher Scientific #13778075) and then transfected again after 24 hours using forward transfection with 20 nM of siRNA. Cells were harvested at either 36 or 48 hours starting

from the initial reverse transfection. For end-point assays requiring fluorescent imaging, cells were grown on coverslips. Detailed protocols on immunofluorescent assays performed on the cultured cells are available in Supplementary Materials. For cell cycle profiling using PI flow cytometry, cells were fixed by 90% chilled ethanol overnight. Fixed cells were washed and resuspended in PBS containing 1% NGS and then incubated with 10 μg/ml of RNase A and 120 μg/ml of propidium iodide (PI) for 30 min in the dark at 37 °C. Samples were analyzed on BD LSR II analyzer. Data were processed using FCS Express version 6 software.

## Statistics and Reproducibility

All quantitative measurements (i.e., weights, serum assays, western blots) have at least three independent biological repeats. The results were expressed with mean and standard deviation unless mentioned otherwise. Differences between groups were examined for statistical significance using unpaired $t$-test with Welch's correction (for two groups), or one-way ANOVA for more than two groups using core functions available in R, an open-source statistical computing software environment. $P$ value <0.05 or FDR < 0.10 was considered significant. All data plots were generated in R using the ggplot2 package. All boxplots are plotted with lower and upper quartiles (Q1 and Q3) corresponding to the 25th and 75th percentile, the median represented by a horizontal line, and whiskers showing the maxima and minima. Outliers in the boxplots are data points with values greater than Q3 + 1.5 x inner-quartile range or less than Q1 − 1.5 x inner-quartile range. Heatmaps were created using the heatmap.2 function, available in the gplots package in R. The gene ontology network map was created using Cytoscape 3.8.0[84]. All western blots and cell culture experiments were repeated independently at least 3 times with similar results.

## Reporting summary

Further information on research design is available in the Nature Portfolio Reporting Summary linked to this article.

## Data availability

The raw RNA-seq data are available for download from NCBI Gene Expression Omnibus under the accession numbers GSE147005 and GSE179634. The mass spectrometry proteomics data were deposited to the ProteomeXchange Consortium via the PRIDE partner repository with the dataset identifier PXD027035. Source data are provided with this paper.

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

## Acknowledgements

We thank the members of the Kalsotra and Anakk laboratories for their valuable discussions and comments on the manuscript. This research was supported through the NIH grants R01AA010154, R01HL126845, and R21HD104039 (to A.K.), R01DK113080 (to S.A.), R01GM132458 and R21AG065748, (to K.V.P.), and R01HG004659, U41HG009889 (to G.W.Y.); the William C. Rose Professorship and the Beckman Fellowship from the Center for Advanced Study at the University of Illinois (to A.K.); Natural Sciences and Engineering Research Council of Canada grant RGPIN-2020-07212 (to C.L.C.); Cancer center @ Illinois seed grants (to A.K, K.V.P, and S.A); the American Cancer Society grant RSG ACS132640 (to S.A.); the NSF grant MCB1723008 (to K.V.P.); the NIH pre-doctoral NRSA fellowship F30DK108567 (to W.A.); the NIH Tissue microenvironment training program T32-EB019944 and UIUC Scott Dissertation Fellowship (to S.B.); and the Ontario Graduate Scholarship (to M.F.S.).

## Author contributions

W.A. and A.K. conceived the project and designed experiments. W.A. performed all experiments and analyzed the data. B.M. and S.A performed qPCRs and assisted with tissue harvesting and animal studies. K.T. assisted with immunofluorescence staining and splice assays. M.F.S. and C.L.C. performed mass spectrometry and differential proteomics analysis. S.M.B., B.A.Y., and E.V.N. performed eCLIP experiments and peak calling analysis. G.W.Y. supervised eCLIP experiments. Y.J.S. performed and K.V.P supervised the PI flow cytometry and tubulin staining experiments. U.V.C., S.B., Q.H. assisted with cell culture, protein synthesis, and western blot experiments. S.K. and G.G. examined and scored all histological specimens. W.A. and A.K. interpreted the results and wrote the manuscript. All authors discussed the results and edited the manuscript.

## Competing interests

G.W.Y is a co-founder, member of the Board of Directors, on the scientific advisory boards, equity holder, and paid consultant for Locanabio and Eclipse BioInnovations. G.W.Y is a visiting professor at the National University of Singapore. G.W.Y's interest(s) have been reviewed and approved by the University of California, San Diego in accordance with its conflict-of-interest policies. E.V.N is co-founder, member of the Board of Directors, on the SAB, equity holder, and paid consultant for Eclipse BioInnovations. E.V.N's interests have been reviewed and approved by the University of California, San Diego in accordance with its conflict of interest policies. Other authors declare no competing interests.
