## [Peer Review File · Nature Communications]

Splicing Factor SRSF1 Deficiency in the Liver Triggers NASH-like Pathology and Cell DeathREVIEWER COMMENTS

Reviewer #1 (Remarks to the Author):

In this manuscript, Arif and colleagues generated mice where the splicing factor *Srsf1* was inactivated selectively in hepatocytes. They have developed two mouse models. In the first one, they used a conventional strategy to generate liver-specific depletion by crossing *Srsf1*^{flox/flox} with an AlbCre transgenic mice. There were some problems since the hepatic knockout of SRSF1 triggered immediate repopulation with SRSF1-expressing hepatocytes. They observed an acute and reversible NASH-like pathology in adult SRSF1 HKO mice. The authors went on to investigate the mechanism of hepatocyte repopulation in SRSF1 HKO livers (discussed below in Specific comments).

Due to the limitations observed with their first *Srsf1* mouse model, the authors pursued the generation of mice with acute hepatocyte-specific knockout of SRSF1. Arif and colleagues conclusively show that acute SRSF1 HKO mice develop hepatic steatosis and liver failure. The authors show that genotoxic stress following SRSF1 depletion leads to inhibition of transcription and protein synthesis, leading to impaired metabolism and trafficking of lipids. An interesting observation is that the acute liver damage observed upon targeted SRSF1 deletion in mice liver is the result of increased formation of R-Loops. There have been some previous studies showing that depletion of individual SR proteins leads to R-loop formation, mainly done in cells in culture (Manley and Fu labs)

The main message of this study is that SRSF1 is essential for hepatocyte function and survival. The authors put forward a model where SRSF1 is essential to preserve genome integrity and tissue homeostasis.

In summary, there is a lot of work, the experiments have been carefully designed and the conclusions are well supported by the data. From a conceptual point of view, there is not a significant mechanistic advance here. In summary, this is a good paper with some relevant findings. My very subjective evaluation of this study is that it is borderline for publication in Nat commun, mostly based on comments outlined below.

Specific comments

1- On page 6 'Surprisingly, despite the drastic liver phenotype, SRSF1 levels in the knockout model at PN10 displayed only a two-fold reduction. This finding was unexpected because previous reports utilizing a Cre-dependent reporter have shown uniform AlbCre transgene activity across all hepatocytes by PN3 (37, 38). Hence, a greater reduction of SRSF1 was anticipated in the knockout model'

This is a general problem with Cre drivers, if they are inefficient the phenotypes can be difficult to analyze/interpret due to there being mixtures of deleted/non-deleted cells in the tissues being studied.

Comparing the efficiency of Alb-Cre deletion at *Srsf1* with that observed in a different paper with a

reporter gene is not great – it is well known that the mouse genetic background can influence this and that different floxed genes/alleles recombine with different efficiencies even with the same Cre. Thus, this finding is only 'unexpected' because they authors have extrapolated from a reporter published elsewhere.

2- 'The greatest decrease in SRSF1 expression within hepatocytes was observed at PN6, which coincided with the onset of damage and steatosis detected histologically. After this time point, SRSF1-deficient hepatocytes were slowly replaced by an expanding population of SRSF1-positive hepatocytes'. We hypothesized that the parenchymal repopulation with SRSF1 expressing hepatocytes resulted either from

Western blot images for PN6 and NOT only for PN10 should be shown in Fig S1B. Why do the authors selected PN10 when they claim PN6 showed the greatest decrease?

3- The authors put forward two models that could lead to SRSF1 repopulation of livers

a) expansion of wildtype hepatocytes that have escaped Cre-mediated SRSF1 knockout

b) and/or continuous transdifferentiation of biliary epithelial cells (BEC) into hepatocytes'

In addition to explanations (a) and (b) is it also possible in their system that Alb-Cre is not uniformly expressed across all hepatocytes? Do the authors have any data for this? In terms of impact, it sounds like the main new finding/message is that SRSF1 is needed in hepatocytes? The repopulation of the liver with Cre-Lox escape is predictable and expected for a Cre-Lox experiment with any gene that is essential/important in hepatocytes (e.g. RNA polymerase floxed alleles would likely show the same thing).

4- On page 9

'Altogether, these findings demonstrate that liver damage in SRSF1 HKO mice occurs in two phases consisting of (1) acute injury which begins immediately after birth and (2) chronic injury that lasts about three months, at which point the liver is completely repopulated with SRSF1-expressing hepatocytes'

Again, this is not biological conclusion per se, rather a reflection of the limitations of the use of AblCre in this system

5- On page 21, the authors state

“Even more astonishing was that SRSF1 HKO mice livers ultimately evaded the deletion and repopulated their parenchyma with SRSF1-expressing hepatocytes”

This is not astonishing per se and probably does not represent an important biological finding but has more to do with the efficiency of the Cre ablation, thus, not many conclusions can be extracted from here.

5- On page 15,

‘Collectively, these findings suggest that the majority of transcriptome defects arising after acute SRSF1 deletion are independent of its splicing activity and likely result from defects in mRNA transcription, export, stability, and/or translation’

The authors do not elucidate a clear mechanistic view of how SRSF1 depletion affects liver biology. The description of phenotypes is of interest, but they cannot clearly identify RNA targets that link their defective processing to the phenotypes observed. In other words, the authors have not defined the molecular events that lead to severe metabolic insufficiency and eventual hepatic failure due to the hepatic loss of SRSF1

Reviewer #2 (Remarks to the Author):

SRSF1 is a crucial splicing factor that regulates various aspects of mRNA metabolism. In this study, Arif et al. tried to examine and elucidate potential mechanisms for liver damage that is caused by the depletion, knockout, of the SRSF1 gene in the mouse hepatocyte. Since the SRSF1 protein knockout is embryonically lethal, they created Cre-mediated hepatocyte-specific knockout mice (SRSF1 HKO mice). SRSF1 HKO mice could survive to adulthood and yet there was significant liver damage from postnatal period to early adult hood while the damage was reversed by repopulation of normal SRSF1 expressing hepatocyte that escaped the Cre-mediated KO. Authors then created acute SRSF1 KO in hepatocyte mice (acSRSF1 HKO mice), to study the primary change in the hepatocyte immediately following the SRSF1 depletion that occur prior to the significant damage in liver. Following transcriptome and proteome wide analysis carried out by the author suggested that SRSF1 depletion caused genome instability which led to mRNA transcription and processing defect that also resulted in significant impairment of protein synthesis resulting in cell death which also resulted in impairment of mouse liver function and NASH-like liver pathology. Furthermore, authors also found that knock down of SRSF1 in human liver cancer cells recapitulated most of the molecular change including genomic instability that they observed in mouse hepatocyte.

Authors generated the mouse system that can provide novel insight into the effect of SRSF1 depletion in mouse hepatocyte and performed extensive array of proteome and transcriptome wide analysis on that

system. Creation of acute KO system distinguished, to some extent, the initial and chronic change that can be brought by the SRSF1 depletion in mouse hepatocyte or other mammalian tissues.

Nonetheless, molecular change in hepatocyte after SRSF1 KO is somewhat expected, including the R-loop formation, and while the authors argue that splicing, which is relatively well known function of SRSF1, has relatively minor consequences and yet I do not necessarily agree with their conclusion there (further explained in my major point). The direct link between SRSF1 and R-loop formation in hepatocyte is missing and in that regard the study failed to provide additional knowledge on the function of SRSF1 in maintaining genome stability. On the similar note, authors failed to further elaborate on the link between SRSF1 and hepatocyte unique gene regulation or liver tissue homeostasis.

On another note, relatively complete reversal of liver damage induced by SRSF1 depletion, despite its extensiveness due to the complete depletion of the SRSF1 gene in hepatocyte, seem to indicate that SRSF1 depletion or mutation is unlikely to be a major cause for the development of human NASH-like liver pathology. Nevertheless, this point may be addressed by the analysis of TCGA or GWAS data to find significant SRSF1 mutations in patients with liver problems.

This study provides significant amount of knowledge and the resource that can be useful for the future study on SRSF1 function in mouse liver and other tissues. However, the manuscript failed to provide the clear evidence that support the main conclusion of this study, i.e., the importance of SRSF1 in maintaining genome stability in relation to human disease. Given that SRSF1 is a vital gene involved in multiple pathways and layers of mRNA metabolism, findings in this study remain rather descriptive. Unless the authors can provide additional evidence or analysis that can clarify the hepatocyte unique function of SRSF1 and/or direct link between the R-loop formation and SRSF1 in hepatocyte, I recommend the paper to be published in a more specialized journal.

Major points

1. Page 15 line 31 “Collectively, these findings suggest that the majority of transcriptome defects arising after acute SRSF1 deletion are independent of its splicing activity and likely result from defects in mRNA transcription, export, stability, and/or translation.” Here, the authors downplayed the importance of splicing change/defect brought by the SRSF1 depletion on the basis of 1) splicing change/defect did not result in significant change in the final transcript level, and 2) SRSF1 eCLIP-seq in hepatocyte did not find significant amount of SRSF1 binding in the transcripts with change in splicing. Does that mean splicing defect is secondary to other damage brought by the SRSF1 depletion? I believe the conclusion drawn here is an overstatement since over 1500 differential splicing event, with or without the change in transcript expression level may have had significant consequences in hepatocyte.

2. Page 17 line 92 “Collectively these data provide compelling evidence that SRSF1 deficiency results in excessive accumulation of R-loops, resulting in subsequent DNA damage and global transcription repression.” The observation after the SRSF1 depletion is in line with the authors’ conclusion here and yet direct link between SRSF1 formation and R-loop formation seem to be still unclear. Could the authors further evaluate distinct nature of R-loop formation after SRSF1 depletion in hepatocyte? Could

there be possibility that R-loop formation is more prevalent in SRSF1 binding gene transcription sites?

3. Fig. 8 and S7 Since the mitochondrial ribosomal proteins are also translated in cytoplasm, it isn't clear how they would escape the global reduction the protein translation level. It is indeed interesting that translation factors and ribosomal proteins seem to be differentially affected. Authors raised the possible involvement of the mTOR pathway, apart from the SRSF1 eCLIP-seq, could they analyze the mTOR sequence enrichment in the transcripts of the downregulated proteins?

4. Given the relatively limited depth of the proteome analysis, little overlap between down regulated gene and DAP may not be compelling evidence for independence of protein abundance and transcript expression level. More in depth proteome analysis and direct system wide comparison of transcriptome and proteome data may be helpful. It is perhaps interesting to note that while the change may not be significant; there seems to be noticeable difference between mitochondrial ribosomal protein transcript level and the other proteins which were downregulated (Fig. 8A).

5. Regarding the mass spectrometry data, label-free quantification must undergo sample-wise normalization (i.e. global normalization or using a reference protein). I am missing description on such normalization procedures both in the main text or the methods section. How were the quantitation values normalized between samples, if any? If the proteomes are largely similar and show good linearity, I suggest using normalized MaxLFQ values instead of iBAQ, for more accurate quantitative values. Also, some imputation approaches for zero-values can improve the statistical power.

6. I wonder whether the authors considered the fact that general poly A transcript level in SRSF1 KO cells are significantly downregulated when they performed the puromycin incorporation assay and polysome profiling (Fig. 8B, S7H). Protein level normalization may skew the puromycin incorporation assay, due to lower amount of transcript present. RNA level normalization in polysome profiling may also result in lower signal of translating mRNA, since the rRNA level is not affected in the SRSF1 KO.

Minor

1. Figs. 1B, 2A, 5C: Please provide images of individual channels. The images are hard to interpret.

2. While the poly A transcript level seem to have decreased significantly, there were greater number of upregulated genes after SRSF1 depletion (Fig 7C and Fig S5B), could there be an explanation on this phenomenon?

Reviewer #3 (Remarks to the Author):

In this manuscript the authors report that loss of SRSF1 in liver causes DNA damage and metabolic derrangements. Overall it is a nice paper with interesting results but there are a few points that need to be addressed. A hepatocyte knockout has been reported previously by Cheng et al. Ref 19. Although the main focus of that paper was SRSF2, the did include SRSF1 knockouts but the results were mainly

negative. The authors need to explain why their results differ from Cheng et al who reported that hepatocyte knockout of SRSF1 did not have a phenotype. Is it a different background for the mice?

In young mice the hepatocytes show deletion of SRSF1 but by 10 days there is evidence for SRSF1 +ve cells, by 5 weeks 50% of cells are +ve, and by 3 months all hepatocytes are SRSF1 +ve. The authors show that Cre expression is repressed similar to what has been reported by others (eg ref 39). But livers from KO mice at 3 and 6 months still show histological changes (inflammation, steatosis, fibrosis) even though SRSF1 is present. Why do changes persist if hepatocytes are not SRSF1 KO?

Fig. 1 oil-red O staining needs quantification from multiple sections from multiple mice. The SRSF1 levels only appear to be reduced 50% in whole liver in fig s1b. The author should check levels in hepatocytes as SRSF1 is expressed in the non-parenchymal cells too. The Ki67 staining needs quantification. Is the SRSF1 KO Ki67 at PN10 really different from control? Is the Ki67 in the SRSF1 +ve or -ve cells? Is this proliferation of the hepatocytes that have repressed Cre expression?

SRSF1 levels are reduced 50% at PN10 but are still reduced 40% at 5 weeks when 50% of the hepatocytes have recovered SRSF1 expression. The data are not consistent. I would have expected a better recovery by western blot.

The authors perform a GTT at 8 weeks and do not see any difference. It would be interesting to know whether younger mice say at 1 month, would show a difference when the SRSF1 KO is more profound. Later on they report that the KO hepatocytes are steatitic, so it would be more informative to look at lipid metabolism rather than glucose metabolism. Is there a change in lipid synthesis or storage?

Fig 3 all histology needs quantification and NASH scoring by a blinded pathologist.

The paper series heavily on bioinformatic predictions. The expression profiling, proteomics and splicing analysis is interesting but is only suggestive of potential changes. These predictions need to be followed up with biochemical or physiological verification. Did they verify any changes in splicing of any genes by PCR? Did they verify the reduction in proteins by western blot?

The authors state that SRSF1 deficient cells do not undergo apoptosis and suggest that cells may be dying by necrosis. What is the evidence that cells are dying? Did they measure apoptosis in the HKO model as well as the AAV8? They state that there was evidence that cells were dying in fig S1c but all they show are H&E and Ki67 staining. There are a few ballooning hepatocytes but nothing showing dying cells. The siSRSF1 in HepG2 cells shows apoptosis so it is surprising they did not detect it in the tissue.

Acute loss of SRSF1 causes liver damage with elevated liver enzymes. The authors state that this leads to eventual liver failure. What is the evidence for liver failure? What is the mortality in these mice? Or is it just reversible liver damage rather than failure?

One of the more interesting aspects of the paper is the DNA damage caused by SRSF1 loss. The authors

confirm the finding from Jim Manley's lab that SRSF1 loss causes R-loops and DSBs. Do they see R-loops at later times? The DNA damage is still observed at 5 weeks. Is this in SRSF1-ve hepatocytes? Do the SRSF1 replenished cells show DNA damage? The authors should look at activation of DNA damage response pathways in response to SRSF1 loss.

Polyploidy through endoreplication and failed cytokinesis is a mechanism to protect from DNA damage. Most hepatocytes are tetraploid either binuclear or mononuclear. Do the SRSF1 KO hepatocytes have a change in ploidy or bi-nuclearity?

What is the mechanism of translational repression? There is no change in eIF2 phosphorylation or expression. What is causing the large scale reduction in translation factor expression?

The authors show that p53 did not mediate the defect in protein synthesis or cell death but they should also examine cell cycle arrest. It is possible that the SRSF1 deficient cells arrest to allow for DNA damage repair. If loss of SRSF1 causes DSBs, then is there a carcinogenic phenotype in these mice? Are they predisposed to HCC?

Nature Communications manuscript NCOMMS-21-29528

Response to Reviewer Comments

We thank the Reviewers for their valuable input and constructive evaluation of our work. We have now performed additional experiments that address the comments made by all three Reviewers. We have also modified and clarified the text in the manuscript accordingly. The new experiments have solidified the conclusions and have improved the quality of the paper. Our point-by-point responses are listed below.

Reviewer #1 Comments:

In this manuscript, Arif and colleagues generated mice where the splicing factor *Srsf1* was inactivated selectively in hepatocytes. They have developed two mouse models. In the first one, they used a conventional strategy to generate liver-specific depletion by crossing *Srsf1*^{flox/flox} with an AlbCre transgenic mice. There were some problems since the hepatic knockout of SRSF1 triggered immediate repopulation with SRSF1-expressing hepatocytes. They observed an acute and reversible NASH-like pathology in adult SRSF1 HKO mice. The authors went on to investigate the mechanism of hepatocyte repopulation in SRSF1 HKO livers (discussed below in Specific comments).

Due to the limitations observed with their first *Srsf1* mouse model, the authors pursued the generation of mice with acute hepatocyte-specific knockout of SRSF1. Arif and colleagues conclusively show that acute SRSF1 HKO mice develop hepatic steatosis and liver failure. The authors show that genotoxic stress following SRSF1 depletion leads to inhibition of transcription and protein synthesis, leading to impaired metabolism and trafficking of lipids. An interesting observation is that the acute liver damage observed upon targeted SRSF1 deletion in mice liver is the result of increased formation of R-Loops. There have been some previous studies showing that depletion of individual SR proteins leads to R-loop formation, mainly done in cells in culture (Manley and Fu labs)

The main message of this study is that SRSF1 is essential for hepatocyte function and survival. The authors put forward a model where SRSF1 is essential to preserve genome integrity and tissue homeostasis. In summary, there is a lot of work, the experiments have been carefully designed and the conclusions are well supported by the data. In summary, this is a good paper with some relevant findings.

Thank you for the positive evaluation of our work. We appreciate the insightful suggestions that have helped us clarify our claims and statements.

1. On page 6 ‘Surprisingly, despite the drastic liver phenotype, SRSF1 levels in the knockout model at PN10 displayed only a two-fold reduction. This finding was unexpected because previous reports utilizing a Cre-dependent reporter have shown uniform AlbCre transgene activity across all hepatocytes by PN3 (37, 38). Hence, a greater reduction of SRSF1 was anticipated in the knockout model’ This is a general problem with Cre drivers, if they are inefficient, the phenotypes can be difficult to analyze/interpret due to there being mixtures of deleted/non-deleted cells in the tissues being studied. Comparing the efficiency of Alb-Cre deletion at *Srsf1* with that observed in a different paper with a reporter gene is not great – it is well known that the mouse genetic background can influence this and that different floxed genes/alleles recombine with different efficiencies even with the same Cre. Thus, this finding is only 'unexpected' because the authors have extrapolated from a reporter published elsewhere.

Thank you for raising this important point. Although it is known that Cre recombination efficiencies can vary among different genetic backgrounds and floxed alleles, we do not believe it is a likely problem with Alb-Cre mice. For instance, in **Fig. 1c**, we show that the number of SRSF1-positive hepatocytes reaches a minimum at post-natal day 6, with an average of 90% knockout. Therefore, the recombination efficiency is significant. However, after this point, the liver begins repopulating with SRSF1-positive hepatocytes.

Furthermore, as shown in **Fig. 2b** of the revised manuscript, the mRNA expression of Cre in SRSF1 HKO mice decreases drastically over time. This indicates that the expanding population of SRSF1-positive hepatocytes in the 5-week and adult livers of SRSF1 HKO mice are deficient in Cre activity. The mechanism of this loss of Cre activity could be due to suppressed gene expression or loss of the DNA locus (PMID: 19272382).

Nonetheless, to address the Reviewer's concern, we performed additional experiments and evaluated the efficiency of Alb-Cre activity by generating hepatocyte-specific knockout of two separate splicing factors (PTBP1 and RBFOX2). In both cases, we achieved efficient and stable knockout through postnatal development and into adulthood. As shown below in our unpublished data, the co-immunofluorescence images demonstrate that hepatocyte expression of PTBP1 decreases progressively and completely in *Ptbp1^{fl/fl}; Alb-Cre^{+/-}* mice relative to *Ptbp1^{fl/fl}* littermate controls (**Response Fig. 1A, B**). Similarly, western blots performed on isolated adult hepatocytes show a near-complete absence of RBFOX2 protein in *Rbfox2^{fl/fl}; Alb-Cre^{+/-}* mice livers relative to *Rbfox2^{fl/fl}* littermate control mice livers (**Response Fig. 1C**).

Response Figure 1: (A). Immunofluorescence images of hepatocyte-specific PTBP1 knockout progression through various stages of postnatal liver development. PTBP1 is present in low amounts at postnatal day 7 and 14, and is absent from hepatocytes of adult *Ptbp1^{fl/fl}; Alb-Cre^{+/-}* mice livers. Immuno-fluorescence images show co-staining with HNF4α (green), PTBP1 (red), and Nuclei (blue). **(B).** Quantification of the percentage of PTBP1-expressing hepatocytes per field from immunofluorescence images at each timepoint shown in A. **(C).** Western blots depicting near-complete loss of RBFOX2 in hepatocytes isolated from adult *Rbfox2^{fl/fl}; Alb-Cre^{+/-}* mice livers. TBP and Ponceau staining were used as loading controls.

2. 'The greatest decrease in SRSF1 expression within hepatocytes was observed at PN6, which coincided with the onset of damage and steatosis detected histologically. After this time point, SRSF1-deficient hepatocytes were slowly replaced by an expanding population of SRSF1-positive hepatocytes. We hypothesized that the parenchymal repopulation with SRSF1 expressing hepatocytes resulted either from'.

Western blot images for PN6 and NOT only for PN10 should be shown in Fig S1B. Why do the authors selected PN10 when they claim PN6 showed the greatest decrease?

The Reviewer is correct that the greatest decrease was observed at PN6, as shown from the SRSF1 immunofluorescence staining in **Fig. 1b**. But PN6 timepoint could not be used for western blot analysis because it is challenging to isolate good-quality hepatocytes at this time. We use the two-step collagenase perfusion technique, which requires perfusion of the liver via cannulation through the portal vein. At PN6, the veins are too small to cannulate. Therefore, the earliest time point at which we were able to isolate hepatocytes was PN10.

3. The authors put forward two models that could lead to SRSF1 repopulation of livers
a) expansion of wildtype hepatocytes that have escaped Cre-mediated SRSF1 knockout
b) and/or continuous transdifferentiation of biliary epithelial cells (BEC) into hepatocytes'

In addition to explanations (a) and (b) is it also possible in their system that Alb-Cre is not uniformly expressed across all hepatocytes? Do the authors have any data for this? In terms of impact, it sounds like the main new finding/message is that SRSF1 is needed in hepatocytes? The repopulation of the liver with Cre-Lox escape is predictable and expected for a Cre-Lox experiment with any gene that is essential/important in hepatocytes (e.g. RNA polymerase floxed alleles would likely show the same thing).

We agree it is possible that Alb-Cre is not expressed uniformly across all hepatocytes, and that is precisely what we meant by our explanation in (a). But, based on our recent results from PTBP1 and RBFOX2 knockouts in hepatocytes, that seems unlikely (see our response to comment 1). In any case, the subpopulation of hepatocytes — that do not express Cre to begin with or shut it down — expands and repopulates the liver over time. Again, the mechanism for the loss of Cre expression is not clear. This could be due to suppression via epigenetic effects, or through loss of the CRE transgene locus, etc. And yes, we concur that a similar phenotype would be seen with the knockout of any essential/important hepatic gene using Alb-Cre recombinase.

4. On page 9, 'Altogether, these findings demonstrate that liver damage in SRSF1 HKO mice occurs in two phases consisting of (1) acute injury which begins immediately after birth and (2) chronic injury that lasts about three months, at which point the liver is completely repopulated with SRSF1-expressing hepatocytes.'

Again, this is not biological conclusion per se, rather a reflection of the limitations of the use of Alb-Cre in this system

We agree with the Reviewer. However, we report these findings here as this might be valuable knowledge for the broad scientific community and raise awareness about the limitations of the Cre-LoxP approach. We have pointed this out in the Discussion section (**Lines 512-525**).

5. On page 21, the authors state, "Even more astonishing was that SRSF1 HKO mice livers ultimately evaded the deletion and repopulated their parenchyma with SRSF1-expressing hepatocytes" This is not astonishing per se and probably does not represent an important biological finding but has

more to do with the efficiency of the Cre ablation, thus, not many conclusions can be extracted from here.

As we have mentioned previously, we believe the repopulation of liver parenchyma in SRSF1 HKO mice with SRSF1-expressing hepatocytes is not due to inefficient Cre activity. Instead, it is a result of the expansion of a population of hepatocytes that have lost the Cre expression. The evidence to support this claim can be seen in **Fig. 2b**, which shows progressive loss of Cre mRNA in hepatocytes over time.

6. On page 15, 'Collectively, these findings suggest that the majority of transcriptome defects arising after acute SRSF1 deletion are independent of its splicing activity and likely result from defects in mRNA transcription, export, stability, and/or translation'. The authors do not elucidate a clear mechanistic view of how SRSF1 depletion affects liver biology. The description of phenotypes is of interest, but they cannot clearly identify RNA targets that link their defective processing to the phenotypes observed. In other words, the authors have not defined the molecular events that lead to severe metabolic insufficiency and eventual hepatic failure due to the hepatic loss of SRSF1.

We apologize for the confusing statement. In the revised manuscript, we have corrected the sentence to state, "Collectively, these findings suggest that the majority of transcriptome defects arising after acute SRSF1 deletion are independent of its splicing activity" (**Lines 340-342**). We believe the large transcriptome defects observed in SRSF1 HKO are secondary to the R-loop accumulation and DNA damage. In our final model, we have defined the chain of events that lead to the observed phenotype due to hepatic loss of SRSF1 (**Fig. 10**), which shows that the widespread DNA damage following SRSF1 depletion results in the expected insufficiency of hepatic proteins that are necessary for liver function and metabolism. A brief mechanistic explanation for the demise of SRSF1-deficient hepatocytes is also provided in the Discussion section (**Lines 540-546**).

Reviewer #2 Comments:

SRSF1 is a crucial splicing factor that regulates various aspects of mRNA metabolism. In this study, Arif et al. tried to examine and elucidate potential mechanisms for liver damage that is caused by the depletion, knockout, of the SRSF1 gene in the mouse hepatocyte. Since the SRSF1 protein knockout is embryonically lethal, they created Cre-mediated hepatocyte-specific knockout mice (SRSF1 HKO mice). SRSF1 HKO mice could survive to adulthood and yet there was significant liver damage from postnatal period to early adult hood while the damage was reversed by repopulation of normal SRSF1 expressing hepatocyte that escaped the Cre-mediated KO. Authors then created acute SRSF1 KO in hepatocyte mice (acSRSF1 HKO mice), to study the primary change in the hepatocyte immediately following the SRSF1 depletion that occur prior to the significant damage in liver. Following transcriptome and proteome wide analysis carried out by the author suggested that SRSF1 depletion caused genome instability which led to mRNA transcription and processing defect that also resulted in significant impairment of protein synthesis resulting in cell death which also resulted in impairment of mouse liver function and NASH-like liver pathology. Furthermore, authors also found that knock down of SRSF1 in human liver cancer cells recapitulated most of the molecular change including genomic instability that they observed in mouse hepatocyte.

Authors generated the mouse system that can provide novel insight into the effect of SRSF1 depletion in mouse hepatocyte and performed extensive array of proteome and transcriptome wide analysis on that system. Creation of acute KO system distinguished, to some extent, the initial and chronic change that can be brought by the SRSF1 depletion in mouse hepatocyte or other mammalian tissues.

This study provides significant amount of knowledge and the resource that can be useful for the future study on SRSF1 function in mouse liver and other tissues. However, the manuscript failed to provide the clear evidence that support the main conclusion of this study, i.e., the importance of SRSF1 in maintaining genome stability in relation to human disease.

We thank the Reviewer for his/her encouraging comments. We appreciate the constructive feedback and suggestions to strengthen our conclusions further.

1. Relatively complete reversal of liver damage induced by SRSF1 depletion, despite its extensiveness due to the complete depletion of the SRSF1 gene in hepatocyte, seem to indicate that SRSF1 depletion or mutation is unlikely to be a major cause for the development of human NASH-like liver pathology. Nevertheless, this point may be addressed by the analysis of TCGA or GWAS data to find significant SRSF1 mutations in patients with liver problems.

We agree with the Reviewer on this point and followed his/her recommendations to examine the TCGA and GWAS datasets. As expected, we did not find significant SRSF1 mutations in human liver disease (**Response Fig. 2**). This further supports our finding that SRSF1 is vital for the viability of all cell types. In fact, SRSF1 is considered to be a proto-oncogene, and it is often upregulated in many cancers (PMID: 24807918, PMID: 22545246).

Response Figure 2: An OncoPrint track depicting alterations of SRSF1 in the Liver Hepatocellular Carcinoma dataset from the TCGA PanCancer Atlas which contains 348 patient samples. About 4% of the samples show alterations, which generally consist of gene amplification or overexpression.

2. Page 15 line 31: “Collectively, these findings suggest that the majority of transcriptome defects arising after acute SRSF1 depletion are independent of its splicing activity and likely result from defects in mRNA transcription, export, stability, and/or translation.” Here, the authors downplayed the importance of splicing change/defect brought by the SRSF1 depletion on the basis of 1) splicing change/defect did not result in significant change in the final transcript level, and 2) SRSF1 eCLIP-seq in hepatocyte did not find significant amount of SRSF1 binding in the transcripts with change in splicing. Does that mean splicing defect is secondary to other damage brought by the SRSF1 depletion? I believe the conclusion drawn here is an overstatement since over 1500 differential splicing event, with or without the change in transcript expression level may have had significant consequences in hepatocyte.

Yes, this is absolutely correct, and we concur with the Reviewer that most splicing defects detected following SRSF1 depletion are secondary to the DNA damage and R-loop accumulation. We make this conclusion because, of the >5,000 SRSF1 binding sites identified, less than 6% showed significant changes in splicing of the associated exon. We also agree that the splicing defects may exacerbate the observed damage. However, they are unlikely the primary contributing factor here. The DNA damage caused due to the accumulation of deleterious R-loops is the main driving force for hepatocyte damage, resulting in many downstream secondary effects.

3. Page 17 line 92: “Collectively these data provide compelling evidence that SRSF1 deficiency results in excessive accumulation of R-loops, resulting in subsequent DNA damage and global transcription repression.” The observation after the SRSF1 depletion is in line with the authors’ conclusion here and yet direct link between SRSF1 formation and R-loop formation seem to be still unclear. Could the

authors further evaluate distinct nature of R-loop formation after SRSF1 depletion in hepatocyte? Could there be possibility that R-loop formation is more prevalent in SRSF1 binding gene transcription sites?

We understand the Reviewer’s curiosity about evaluating the distinct nature of R-loop formation after SRSF1 depletion. As pointed out by the Reviewer, one would expect to find more R-loop formation on genes that are highly expressed in hepatocytes and to be more prevalent on transcripts that exhibit direct SRSF1 binding. These are exciting prospects but will require a dedicated follow-up study — e.g., genome-wide profiling of R-loops in wildtype and SRSF1-depleted livers.

- Fig. 8 and S7: Since the mitochondrial ribosomal proteins are also translated in the cytoplasm, it isn’t clear how they would escape the global reduction the protein translation level. It is indeed interesting that translation factors and ribosomal proteins seem to be differentially affected. Authors raised the possible involvement of the mTOR pathway, apart from the SRSF1 eCLIP-seq, could they analyze the mTOR sequence enrichment in the transcripts of the downregulated proteins?

The Reviewer raises an interesting point. However, we would like to emphasize that the proteomics data provides relative changes in abundance. We believe that absolute levels of all proteins are diminishing as a result of the downstream effects of DNA damage and reduced transcription. Although we cannot prove this, it appears that the mitochondrial proteins may exhibit longer half-lives compared to other proteins. Therefore, regarding relative abundance, mitochondrial ribosomal proteins do not show a strong downregulation like other ribosomal proteins. As the reviewer points out, the mTOR pathway is a master regulator of cell growth and metabolism. It has kinase activity and can directly impact translation. However, it also has multiple other functions and can indirectly regulate transcription. We examined the expression of genes regulated by transcription factors that are known targets of mTOR (PMID: 23641065). One would expect these genes to be downregulated if SRSF1 deficiency is impairing the activity of mTOR. But, the heatmap in **Response Fig. 3** shows no significant change in mRNA abundance of mTOR target genes.

Response Figure 3: A heatmap of normalized read counts of genes that are indirect targets of mTOR. Genes highlighted in red and blue denote significant upregulation and downregulation, respectively.

5. Given the relatively limited depth of the proteome analysis, little overlap between down regulated gene and DAP may not be compelling evidence for independence of protein abundance and transcript expression level. More in depth proteome analysis and direct system wide comparison of transcriptome and proteome data may be helpful. It is perhaps interesting to note that while the change may not be significant; there seems to be noticeable difference between mitochondrial ribosomal protein transcript level and the other proteins which were downregulated (Fig. 8A).

We apologize that this was not clear in the original manuscript. We have performed a more in-depth analysis comparing the proteomic changes with the transcriptome changes in the acute SRSF1 knockout. This analysis can be found in **Supplementary Fig. 9a-g**. As expected, genes with greater transcript abundance also showed greater protein abundance and *vice versa*. This result provides confidence in the validity of the experiment. However, transcripts that exhibited differential splicing or direct SRSF1 binding did not show any significant effects on the protein abundance.

6. Regarding the mass spectrometry data, label-free quantification must undergo sample-wise normalization (i.e. global normalization or using a reference protein). I am missing description on such normalization procedures both in the main text or the methods section. How were the quantitation values normalized between samples, if any? If the proteomes are largely similar and show good linearity, I suggest using normalized MaxLFQ values instead of iBAQ, for more accurate quantitative values. Also, some imputation approaches for zero-values can improve the statistical power.

We appreciate the Reviewer's expertise on the subject and have clarified the method that we used explicitly in the Materials and Methods section as follows:

“Global proteomics analysis by mass spectrometry. Samples were processed for proteomics according to the FASP protocol and desalted on an in-house prepared C18 tips (73). The peptides were resuspended to 0.5 ug/uL and 1ug was separated on an EASY Spray C18 column (50 cm x 75 cm, 2 µm particle size) (ThermoFisherScientific, Toronto) using an EASY nLC-1200. The mobile phase was composed of 0.1% formic acid in water (A) and 90% acetonitrile with 0.1% formic acid (B). The gradient was as follows: 5 -40% B (0 – 120 minutes), 40% - 100% B (120 – 125 minutes), 100% B (125 – 135 minutes). The peptides were analyzed on a Thermo QExactive HF mass spectrometer in a Top 20 data dependent acquisition mode.”

We did choose iBAQ over MaxLFQ, and while the reviewer is correct that no global normalization took place, we did normalize the input for each sample by quantifying our tryptic peptides using a colorimetric BCA assay and diluting all the samples to 0.5 mg/mL to ensure equal loading of 1 ug on the column. The MaxLFQ algorithm is beneficial for normalizing a dataset allowing for more accurate quantitative values, however it is plagued with stringent requirements resulting in fewer proteins for analysis. The iBAQ algorithm on the other hand, while not globally normalizing the data, does provide more proteins due to its less stringent peptide requirements while maintaining other useful information such as the protein's relative abundance. We chose to use iBAQ because of these advantages after performing some quality control on the results and recognizing that our data did not appear to require normalization. For example, we found no significant variation in the total iBAQ signal across samples (see **Response Fig. 4A, B**). As mentioned by the reviewer, normalization can be done using a “housekeeping” protein. We assessed the amount of histones per sample, similar to that suggested by Matthias Mann's group in the concept of the “proteomic ruler” [[10.1074/mcp.M113.037309](https://doi.org/10.1074/mcp.M113.037309)], and found no difference in the total amount of histones per sample, suggesting equal loading (**Response Fig. 4C**).

Response Figure 4: (A) The sum total of the iBAQ values for each sample. The columns in red represent wildtype and the columns in blue represent SRSF1 KO hepatocytes. (B) A violin plot of all the iBAQ values from each sample showing the distribution of values (no statistically significant difference between samples). (C) The summed total of the iBAQ values for all 9 detected histone proteins in the sample. (D) Comparison of the log2 fold change of SRSF1 HKO vs wildtype using the LFQ (y-axis) or iBAQ (x-axis) values for proteins in Fig. 9a of the original submission. The plot shows the proteins that had non-zero values for all LFQ and iBAQ values. (E) The p-values associated with the comparisons from D. Values falling in the red rectangle maintain significance.

In addition, we compared the LFQ and iBAQ fold changes for the proteins shown in Fig. 9a. We performed this comparison between proteins that had non-zero readings for both the LFQ and the iBAQ values. The result showed that the direction of the fold change was universally consistent, with the iBAQ estimating larger magnitudes. When we compared the fold changes for significance, regardless of whether iBAQ or MaxLFQ was used, we found that the large majority did retain significance (Response Fig. 4D, E). Taken together, we decided to use iBAQ since the LFQ values would have resulted in fewer proteins while not significantly affecting their overall significance.

7. I wonder whether the authors considered the fact that general poly A transcript level in SRSF1 KO cells are significantly downregulated when they performed the puromycin incorporation assay and polysome profiling (Fig. 8B, S7H). Protein level normalization may skew the puromycin incorporation assay, due to lower amount of transcript present. RNA level normalization in polysome profiling may also result in lower signal of translating mRNA, since the rRNA level is not affected in the SRSF1 KO.

This is a valid point and an insightful consideration. We agree that decreased polyA transcript levels in SRSF1 KO hepatocytes greatly contribute to the observed decrease in global protein synthesis, likely exacerbated due to diminishing levels of ribosomal proteins. We have included this point in the Discussion section of the revised manuscript (Lines 540-544).

While we recognize that polysome profiling does not reflect changes in translation rates, our aim to show this result is to provide evidence that the SRSF1 KO hepatocytes exhibit diminished protein synthesis and overall protein output. Therefore, these hepatocytes are deficient in synthesizing necessary proteins required for hepatic metabolism and function, which explains the liver failure-like phenotypes (steatosis, reduced serum cholesterol, reduced fasting ketones, and blood glucose levels) of SRSF1 HKO mice.

8. While the poly A transcript level seem to have decreased significantly, there were greater number of upregulated genes after SRSF1 depletion (Fig 7C and Fig S5B), could there be an explanation on this phenomenon?

We thank the Reviewer for raising this important question. Our explanation for the large number of upregulated genes identified by RNA-Seq analysis despite the global decrease in polyA transcripts is that RNA-Seq measures the relative abundance of transcripts.

Although the overall levels of all transcripts are decreased in SRSF1 HKO, many transcripts display higher relative abundance. For example, let us take a hypothetical case where we measure the abundance of transcripts A, B, and C between control and experimental conditions. In the control sample, the absolute abundance of transcripts A, B, and C are 500, 100, and 400. But in the experimental condition, the absolute abundance of A, B, and C are 10, 50, and 40. As is evident, the absolute abundance of all transcripts has decreased between the two conditions. However, from RNA-Seq analysis, one would find that transcript A is downregulated (from 50% to 10%), transcript B is upregulated (from 10% to 50%), and transcript C is unchanged.

Reviewer #3 Comments:

In this manuscript, the authors report that loss of SRSF1 in liver causes DNA damage and metabolic derangements. Overall, it is a nice paper with interesting results but there are a few points that need to be addressed.

We are gratified by the interest and comments of the Reviewer. We thank him/her for appreciating our work and providing thoughtful suggestions and feedback.

1. A hepatocyte knockout has been reported previously by Cheng et al. Ref 19. Although the main focus of that paper was SRSF2, they did include SRSF1 knockouts but the results were mainly negative. The authors need to explain why their results differ from Cheng et al. who reported that hepatocyte knockout of SRSF1 did not have a phenotype. Is it a different background for the mice?

We appreciate the Reviewer's comment and acknowledge the prior publication of the SRSF1 knockout in Ref. 19. Actually, the findings reported by Cheng et al. are in agreement with our results. In their manuscript, Cheng et al. showed that the survival of SRSF1 HKO is unchanged in comparison to the control, which is what we observe. They also presented H&E stainings of SRSF1 HKO livers at two months which showed minimal difference compared to wildtype livers. Again, this is expected because, by this age, the liver has been essentially repopulated with SRSF1-expressing hepatocytes (see **Figs. 1, 2**). Furthermore, on closer inspection of the H&E images shown by Cheng et al., it is apparent that the hepatocyte nuclei in the SRSF1 HKO were enlarged relative to the wildtype controls, which is consistent with our findings as well (see **Fig. 3**).

2. In young mice, the hepatocytes show deletion of SRSF1 but by 10 days there is evidence for SRSF1 +ve cells, by 5 weeks 50% of cells are SRSF1 +ve, and by 3 months all hepatocytes are SRSF1 +ve. The authors show that Cre expression is repressed similar to what has been reported by others (e.g. ref

39). But livers from KO mice at 3 and 6 months still show histological changes (inflammation, steatosis, fibrosis) even though SRSF1 is present. Why do changes persist if hepatocytes are not SRSF1 KO?

The Reviewer brings up an interesting point. We believe these changes persist for 3 to 6 months because it takes time for the liver to recuperate from the severe liver damage and cell death triggered after SRSF1 depletion. This is expected as the injured liver undergoes tissue remodeling/repair and clears the damaged/dying cells. From our timed histological and serum analyses of SRSF1 HKOs (**Fig. 3; Supplementary Tables 1-3**), it is apparent that steatosis, inflammation, and markers of liver damage (ALT and AST) all normalize with age.

3. The SRSF1 levels only appear to be reduced 50% in whole liver in fig s1b. The author should check levels in hepatocytes as SRSF1 is expressed in the non-parenchymal cells too. The Ki67 staining needs quantification. Is the SRSF1 KO Ki67 at PN10 really different from control? Is the Ki67 in the SRSF1 +ve or -ve cells? Is this proliferation of the hepatocytes that have repressed Cre expression?

We apologize that this was not clear in the original manuscript. The western blot presented in **Supplementary Fig. 1b** was actually performed using isolated hepatocytes, and the Ki67 staining is quantified in **Supplementary Fig. 1d**. The increased Ki67 staining in hepatocytes at PN10 of SRSF1 HKO is further supported by IF staining (**Response Fig. 5**). We hypothesized Ki67 staining to be present in SRSF1-positive hepatocytes, which would also be the cells where Cre expression is repressed. To test this hypothesis, we performed co-staining of Ki67 and SRSF1 at the 1-month time point. As expected, the staining shows that Ki67 is present in SRSF1-positive hepatocytes.

Response Figure 5: Hepatic proliferation in SRSF1 HKO mice is limited to SRSF1-positive hepatocytes. Representative IF images of SRSF1 and Ki67 co-staining in 1-month Control and SRSF1 HKO mice (n = 3). In addition to increased proliferation in SRSF1 HKO livers as noted by the greater number of Ki67-positive hepatocytes, the proliferation is limited to SRSF1-positive hepatocytes.

- SRSF1 levels are reduced 50% at PN10 but are still reduced 40% at 5 weeks when 50% of the hepatocytes have recovered SRSF1 expression. The data are not consistent. I would have expected a better recovery by western blot.

Thank you for your keen observation. We believe this degree of inconsistency can be expected due to several factors when comparing quantification based on western blotting and/or immunofluorescence.

Firstly, with IFs, we can only quantify the fraction of positive cells. This is because integrating signal intensity is unreliable as the staining is performed on thin cross-sections.

Secondly, western blot analysis quantifies protein abundance relative to the control samples. The inconsistency in the expected recovery over time is likely due to changes in SRSF1 levels with age.

- The authors perform a GTT at 8 weeks and do not see any difference. It would be interesting to know whether younger mice say at 1 month, would show a difference when the SRSF1 KO is more profound. Later on they report that the KO hepatocytes are steatotic, so it would be more informative to look at lipid metabolism rather than glucose metabolism. Is there a change in lipid synthesis or storage?

We agree with the Reviewer that performing a GTT at an earlier time point could possibly show a more profound difference in the SRSF1 HKO mice. However, we did not perform a GTT at that timepoint because fasting glucose levels were not significantly different at 1 month (see **Supplementary Table 3**). We believe that by 1 month, the SRSF1 HKO livers have enough functional capacity to maintain glucose metabolism. We expect that we would have to perform the GTT at an even earlier time point to observe a significant difference. In line with this expectation, acSRSF1 HKO at 4 weeks does show significant hypoglycemia, indicating that loss of SRSF1 can lead to impaired glucose metabolism.

To address the changes in lipid synthesis and storage, we examined expression data for lipid metabolic genes at the 1-month time point (**Response Fig. 6**). We see that several genes involved in lipid synthesis, catabolism, and storage change in their expression pattern. This is likely a secondary response to the lipid build-up in the tissue. However, it should be noted that interpretation of this data is complicated because, at this time point, there is a mixed population of SRSF1-negative and -positive hepatocytes.

Response Figure 6: Lipid metabolic genes changing in expression in 1-month SRSF1 HKO. Bar plots of fold changes in the expression of genes involved in lipid biosynthesis, catabolism, and storage. The list of genes were obtained from gene ontology annotations available from the Mouse Genome Database at the Mouse Genome Informatics website (<http://www.informatics.jax.org>). The legends list the number and fraction of genes in each group (no significant change, upregulated, or downregulated). The table on the right lists the gene names and fold change values of the significantly ($|\text{Log}_2(\text{Fold Change})| > 1$ and $\text{FDR} < 0.05$) changing genes.

6. Fig 3 all histology needs quantification and NASH scoring by a blinded pathologist.

As suggested by the Reviewer, all histological slides have now been independently scored by two blinded pathologists. The Kleiner histological NAFLD activity scoring (NAS) system was used. These new quantifications have been included in the revised manuscript (**Supplementary Table 2, shown below**). As expected, the NAS scores reflect that with time, inflammation and steatosis in the transgenic mice improve while fibrosis continues to develop.

Mouse Model	Age	Group	Gender	Steatosis (0-3)	Lobular Inflammation (0-3)	Hepatocellular Ballooning (0-2)	Total NAS Score	Fibrosis Score (0-4)
Transgenic SRSF1 HKO	1 Month	Control	Male	1	0	1	2	0
		Control	Male	0	0	1	1	1
		Control	Male	0	1	0	1	1
		Control	Male	0	0	1	1	2
		Control	Female	0	1	0	1	1
		Control	Female	0	1	0	1	2
		Control	Female	1	1	2	4	2
		Control	Female	0	1	1	2	2
		Average		0.25	0.625	0.75	1.625	1.375
		SRSF1 HKO	Male	2	2	2	6	4
		SRSF1 HKO	Male	2	1	2	5	3
		SRSF1 HKO	Male	3	1	2	6	4
		SRSF1 HKO	Male	2	2	2	6	4
		SRSF1 HKO	Female	1	2	2	5	4
	SRSF1 HKO	Female	1	2	2	5	4	
	SRSF1 HKO	Female	1	2	2	5	4	
	SRSF1 HKO	Female	2	2	2	6	3	
	Average		1.75	1.75	2	5.5	3.75	
	Δ Score		+1.5	+1.125	+1.25	+3.875	+2.375	
	3 Month	Control	Male	0	1	0	1	1
		Control	Male	0	1	0	1	1
		Control	Male	0	1	2	3	2
		Control	Male	0	1	2	3	1
		Control	Female	0	1	0	1	1
		Control	Female	0	0	2	2	1
		Average		0.00	0.83	1.00	1.83	1.17
		SRSF1 HKO	Male	0	1	1	2	3
		SRSF1 HKO	Male	1	1	2	4	4
		SRSF1 HKO	Male	0	1	2	3	4
		SRSF1 HKO	Male	0	2	2	4	4
		SRSF1 HKO	Male	1	1	2	4	3
		SRSF1 HKO	Female	0	2	2	4	4
		SRSF1 HKO	Female	0	1	2	3	3
	Average		0.29	1.29	1.86	3.43	3.57	
	Δ Score		+0.28	+0.45	+0.86	+1.60	+2.40	
	6 Month	Control	Male	0	0	1	1	1
		Control	Male	0	1	1	2	2
		Control	Male	0	1	1	2	2
		Control	Male	0	1	1	2	1
		Control	Female	0	1	1	2	2
		Control	Female	0	1	1	2	2
		Average		0.00	0.83	1.00	1.83	1.67
SRSF1 HKO		Male	2	1	2	5	4	
SRSF1 HKO		Male	0	1	2	3	3	
SRSF1 HKO		Male	1	2	2	5	4	
SRSF1 HKO		Male	1	0	2	3	3	
SRSF1 HKO		Female	1	1	1	3	3	
SRSF1 HKO		Female	0	1	1	2	4	
SRSF1 HKO		Female	2	1	2	5	4	
SRSF1 HKO	Female	0	1	1	2	4		
Average		0.88	1.00	1.63	3.50	3.63		
Δ Score		+0.88	+0.17	+0.63	+1.67	+1.96		
Acute SRSF1 HKO, viral induction model	2 Week	Control (GFP)	Female	0	0	0	0	0
		Control	Male	0	0	1	1	0
		Control	Male	0	0	1	1	0
		Control	Male	0	1	1	2	1
		Average		0	0.25	0.75	1	0.25
		acSRSF1 HKO	Male	0	1	2	3	1
	acSRSF1 HKO	Male	1	1	2	4	1	
	acSRSF1 HKO	Male	1	2	0	3	1	
	acSRSF1 HKO	Female	0	0	1	1	1	
	Average		0.5	1	1.25	2.75	1	
	Δ Score		+0.5	+0.75	+0.50	+1.75	+0.75	
	4 Week	Control	Female	0	1	1	2	1
		Control	Male	0	1	1	2	1
		Control	Female	0	1	1	2	1
		Control	Female	0	0	0	0	1
		Control	Male	0	1	0	1	1
		Average		0	0.8	0.6	1.4	1
		acSRSF1 HKO	Male	3	2	1	6	2
		acSRSF1 HKO	Male	2	2	2	6	2
		acSRSF1 HKO	Male	1	2	1	4	1
		acSRSF1 HKO	Female	0	2	1	3	0
		acSRSF1 HKO	Male	3	2	1	6	2
acSRSF1 HKO		Male	3	2	1	6	2	
Average			2.00	2.00	1.17	5.17	1.50	
Δ Score			+2.00	+1.20	+0.57	+3.77	+0.50	

Specifically, the average difference in NAS scores between 1-month and 3-month SRSF1 HKO and Control mice are +3.88 and +1.60, respectively. The decrease in the difference shows that the steatosis and inflammation have reduced. However, the difference in fibrosis scoring at 1 and 3 months are +2.38 and +2.40, signifying no improvement.

7. The paper relies heavily on bioinformatic predictions. The expression profiling, proteomics and splicing analysis is interesting but is only suggestive of potential changes. These predictions need to be followed up with biochemical or physiological verification. Did they verify any changes in splicing of any genes by PCR? Did they verify the reduction in proteins by western blot?

Thank you for this suggestion. We have now validated several splicing changes by RT-PCR and changes in protein abundance by western blots. These new validation results have been included in the revised manuscript (**Supplementary Fig. 6, also shown below**). Specifically, we validated skipped exon events in the following genes; *Arhgef19*, *C8a*, *Fdxr*, *Gak*, *Mff*, *Pdk2*, *Pcmd1*, *Pkp4*, and *Sh3glb2*. These genes were sampled from the rMATS output to represent events that are increasing and decreasing in inclusion. It can be seen in **Supplementary Fig. 6a, 6b** that the rMATS estimation of Δ PSI values strongly correlate with the measured Δ PSI by RT-PCR. Similarly, we validated the proteome data by measuring the change in abundance of RPS16 (**Supplementary Fig. 6c**). Western blot measurements showed a fold change of -1.28, while proteomics data estimated a change of -2.13. This variability between the estimation and measurement is within the expected range. Overall, the direction of change for all splicing events and protein abundance is consistent with the changes estimated by the profiling studies.

Supplementary Figure 6. Validation of alternative splicing and proteome data. (a) RT-PCR validation of alternative splicing in transcripts sampled from the rMATs output to represent events that are increasing and decreasing in inclusion between control and acSRSF1 HKO hepatocytes. The bands corresponding to (+) indicate exon inclusion and (-) indicates exon exclusion. n=4. (b) Scatter plot showing comparison of RT-PCR and RNA-seq based Δ PSI values. (c) Western blots showing SRSF1 and RPS16 protein levels in hepatocytes from controls and acSRSF1 HKO mice with TBP serving as loading control. n=4. Comparison of fold changes in protein abundance of SRSF1 and RPS16 based on calculated IBAQ values from mass spectrometry and western blot analysis.

8. The authors state that SRSF1 deficient cells do not undergo apoptosis and suggest that cells may be dying by necrosis. What is the evidence that cells are dying? Did they measure apoptosis in the HKO model as well as the AAV8? They state that cells were dying in fig S1c but all they show are H&E and Ki67 staining. There are a few ballooning hepatocytes but nothing showing dying cells. The siSRSF1 in HepG2 cells shows apoptosis so it is surprising they did not detect it in the tissue.

We appreciate the Reviewer's concern. We have performed TUNEL staining on both the HKO as well as the acute KO models (**Supplementary Fig. 4b**). In either case, we do not observe any TUNEL-positive cells. However, both BAX (apoptotic marker) and RIPK1 (necroptosis marker) are upregulated, as seen from the western blot shown in **Fig. 8c**. Therefore, we believe these hepatocytes must be undergoing necroptosis. To assess if this was the case, we performed Annexin V staining in the HepG2 siSRSF1 knockdown experiments (**Fig. 9a**). Annexin V, a cell death marker, binds to phosphatidylserine when it is present on the outer leaflet of the plasma membrane. Therefore, positive Annexin V staining can be seen in both necroptosis and apoptosis. However, because we observe diffuse Annexin V staining on the membrane with no nuclear blebbing (seen in apoptosis), we conclude that loss of SRSF1 results in necroptotic cell death. We believe that the SRSF1 deficient cells are unable to mount a proper apoptotic response because the loss of SRSF1 results in global impairment of transcription and protein synthesis. Furthermore, the pathologists have found necrotic hepatocytes in 1-month SRSF1 HKO mice. Below in **Response Fig. 7** are H&E images from liver sections of three different SRSF1 HKO mice.

Response Figure 7. Necrotic hepatocytes in SRSF1 HKO mice. Histology images (400x field) from 1-month SRSF1 HKO mice which show necrotic hepatocytes.

9. Acute loss of SRSF1 causes liver damage with elevated liver enzymes. The authors state that this leads to eventual liver failure. What is the evidence for liver failure? What is the mortality in these mice? Or is it just reversible liver damage rather than failure?

We apologize for erroneously describing the acute liver damage as liver failure. We have corrected this in the manuscript (**Lines 55, 268, 532, and 1138**). All mice survive the acute loss of SRSF1 in both models because the livers eventually repopulate the parenchyma with SRSF1-expressing hepatocytes. Therefore,

liver damage is reversible in the context of our model systems. However, if SRSF1 knockout could be maintained in the hepatocytes indefinitely, we would expect all mice to exhibit liver failure and death.

10. One of the more interesting aspects of the paper is the DNA damage caused by SRSF1 loss. The authors confirm the finding from Jim Manley's lab that SRSF1 loss causes R-loops and DSBs. Do they see R-loops at later times? The DNA damage is still observed at 5 weeks. Is this in SRSF1-ve hepatocytes? Do the SRSF1 replenished cells show DNA damage? The authors should look at activation of DNA damage response pathways in response to SRSF1 loss.

The Reviewer makes a valid criticism, and we appreciate his/her detailed and insightful suggestions. Due to antibody incompatibilities, we were unable to perform SRSF1 and γ H2A.x co-staining previously.

We have now acquired the antibodies to allow for the co-staining. Below is a representative image of the co-staining on liver sections from 1-month SRSF1 HKO mice. At 1-month, the SRSF1 HKO mice have a mixed population of SRSF1-deficient and SRSF1-positive hepatocytes. It is evident from the co-IF that γ H2A.x is present only in SRSF1-negative hepatocytes. We did not find any γ H2A.x in SRSF1-positive hepatocytes. These new data are now included in the revised manuscript (**Supplementary Fig. 8**).

Supplementary Figure 8. Loss of SRSF1 results in DNA damage. Representative immunofluorescence (IF) images of liver sections from 1-month SRSF1 HKO mice ($n = 3$) probed for SRSF1 (green), γ H2A.x (red), a DNA damage marker, with nuclear counterstaining using DAPI (blue).

We would also like to clarify the Reviewer's use of the term 'SRSF1 replenished' cells. We do not believe that SRSF1-deficient hepatocytes revert and start re-expressing SRSF1. Our results indicate that SRSF1-deficient hepatocytes undergo death, and the tissue is gradually replaced by SRSF1-positive hepatocytes that never lost SRSF1 to begin with.

11. Polyploidy through endoreplication and failed cytokinesis is a mechanism to protect from DNA damage. Most hepatocytes are tetraploid either binuclear or mononuclear. Do the SRSF1 KO hepatocytes have a change in ploidy or bi-nuclearity?

Thank you for raising this point. We hypothesized that the ploidy/nuclearity of hepatocytes in SRSF1 HKO livers would be lower relative to wildtype livers. This is because acute damage triggers a regenerative/proliferative response in Cre-repressed hepatocytes. It is known that in the process of hepatocyte proliferation, the cells reduce their degree of ploidy/nuclearity to replenish the lost cell population (PMID: 20861837, PMID: 32242122); and the proliferation rates of diploid hepatocytes are higher than that of polyploid hepatocytes (PMID: 30244478). We quantified the nuclearity of hepatocytes in the liver sections from SRSF1 HKO and control mice, and as expected, the nuclearity was significantly

decreased in the SRSF1 HKO livers (**Response Fig. 8A, B**). Likewise, primary hepatocytes isolated from SRSF1 HKO mice showed a marked decrease in binucleation with majority of cells containing a single enlarged nucleus (**Response Fig. 8C**), which is also apparent in the tissue sections (**Response Fig. 8A**). Together, these data demonstrate a clear reduction in cellular ploidy (number of nuclei per cell) in SRSF1 HKO hepatocytes. Whether these mononuclear hepatocytes also undergo a change in molecular ploidy (DNA content per nuclei) remains to be seen.

Response Figure 8. Reduced hepatocyte nuclearity in SRSF1 HKO livers. (A) 40X hematoxylin and eosin images of 3 month Alb-Cre^{+/-} and Srsf1^{fl/fl}; Alb-Cre^{+/-} livers. Binucleated cells are marked by arrows. (B) Quantification of binucleated cells from hematoxylin and eosin images. n=3 animals per group. (C) Images of primary adult hepatocytes isolated from Alb-Cre^{+/-} and Srsf1^{fl/fl}; Alb-Cre^{+/-} mice livers. Similar to the histological tissue sections, SRSF1 HKO primary hepatocytes also display a clear decrease in binucleated cells with enlarged nuclei and larger cell area as compared to controls.

12. What is the mechanism of translational repression? There is no change in eIF2 phosphorylation or expression. What is causing the large scale reduction in translation factor expression?

Thanks for pointing this out. We believe that the striking reduction in protein synthesis (**Fig. 8b**) is primarily due to a decrease in transcription because of DNA damage and R-loop accumulation. The reduction of steady-state polyA mRNA is shown in **Fig. 7d**. While mechanisms of active translational repression in response to DNA damage may be involved, it is not the major contributor. Most known mechanisms of global translational repression result in phosphorylation of eIF2alpha, which is not seen in the case of SRSF1 knockout. These explanations are provided in the revised manuscript's Results and Discussion sections.

13. The authors show that p53 did not mediate the defect in protein synthesis or cell death but they should also examine cell cycle arrest. It is possible that the SRSF1 deficient cells arrest to allow for DNA

damage repair. If loss of SRSF1 causes DSBs, then is there a carcinogenic phenotype in these mice? Are they predisposed to HCC?

We are grateful to the Reviewer for making this thoughtful suggestion. Examining cell cycle arrest in SRSF1-deficient cells was a logical next step proposed by the Reviewer. To investigate this possibility in our SRSF1 knockdowns, we performed cell cycle analysis using flow cytometry with propidium iodide staining. We also carried out this experiment in both HepG2 and HeLa cells to determine the generalizability of the finding. The results unequivocally show that SRSF1 depletion results in a dramatic G2/M arrest. Arrest at the G2/M phase is indicative of DNA damage accumulating during replication which are beyond repair. These new data are now included in the revised manuscript (**Fig. 9b, also shown below**) and result section (**Lines 486-490**). Regarding the question about carcinogenesis and/or HCC predisposition, although the loss of SRSF1 leads to DSBs, the mice do not develop a carcinogenic phenotype or HCC. This is because SRSF1-deficient hepatocytes undergo death and are replaced by healthy SRSF1-positive hepatocytes over time.

b.

Figure 9. (b) Distributions of cell cycle phase determined by PI flow in HepG2 cells treated with the indicated siRNA for 72 hours. Quantification on the right represent the average from n = 3 replicates.

REVIEWER COMMENTS

Reviewer #2 (Remarks to the Author):

The authors have done extensive work to address the reviewers' points. While most of the points had been satisfactorily addressed, some points need further clarification. The numberings are the same as in the original review.

1. Phenotypic outcomes of essential gene KO are meaningful if clinical associations exist. Are there any clinical circumstances where SRSF1 loss (not upregulation) leads to liver diseases (not random cancer)? If the authors cannot spot global trends in GWAS or TCGA data, they can at least find and mention a number of relevant case studies (if any) in the introduction.

2. I think the author's rebuttal here is rather misleading. The focus should be on "how many and which genes" (not the SRSF1 binding sites nor the fractions) are incorrectly spliced upon SRSF1 KO. Can the authors provide the number of genes with splicing defects and their GO analysis results in the supplementary figures? If the numbers reach hundreds and the genes have DNA damage-related roles, it would be rather hasty to rule out the splicing effects as 'secondary': In this case, the authors should at least clearly describe this limitation in the discussion section.

3. I thought this point, i.e., 'R-Loop Induced DNA Damage' was the main argument of this paper, as best represented in the title "Splicing Factor SRSF1 Deficiency in the Liver Triggers NASH-like Pathology *via* R-Loop Induced DNA Damage *and* Cell Death". If the authors are reluctant to address this point in the current manuscript, I suggest the authors remove the italics part from the title and tone down the respective statements throughout the manuscript.

Reviewer #3 (Remarks to the Author):

The authors have made a number of changes to the manuscript that supports their conclusions. There are still a few issues that I think need to be addressed.

1. I thank the authors for including blinded scoring of the NASH histology. It raises some interesting questions. Firstly, the NAS scores need to be analyzed statistically before the authors can say that there is a difference in any score over time. These are ordinal scores, so normal t-tests cannot be used. A non-parametric test or chi-squared might work. Secondly, the authors state that inflammation, ballooning and steatosis decrease but fibrosis does not. Do the authors have an explanation of why the fibrosis does not decrease? Also the Supplementary Table is at odds with the histology shown in Fig. 3c. The authors state that only slight bridging fibrosis remains at 6 months, but that's not what the histologist scores say.

2. It is difficult to compare transcription changes with changes in splicing. The transcriptional analysis

sums all reads across a transcript but splicing only looks at exon junction spanning reads. So the number of reads at each junction is very much lower than the number of reads per transcript. Indeed splicing junction data is very sparse. The number of events is much higher too, up to 10-20 fold more splicing events than genes. This has a big effect on the statistical analysis when multiple correction is applied as the n is much larger. It is often helpful to limit the analysis of splicing events to those observed in all of, or the majority of, the samples. This reduces the n considerably. The analysis of splicing can give very different results if when different approaches are used. The authors should analyze the data using a different approach from rMATS to see if they get the same results.

3. Gene ontology analysis with alternative splicing is also tricky as in most cases it is not known what functional effect the events has (unlike transcription where genes go up and down). So any conclusions from such an analysis should be taken with a grain of salt.

4. The authors included cell cycle analysis as requested and show that siSRSF1 leads to G2/M arrest. Leading to the idea that DNA damage is causing arrest. It would be nice to show that the DNA damage is detected in the arrested cells by co-staining.

5. I think the mechanistic question is still unanswered. How does loss of SRSF1 lead to increased R-loops. Is it related to binding of SRSF1 to the Pol 2CTD?

Reviewer #2 (Remarks to the Author):

The authors have done extensive work to address the reviewers' points. While most of the points have been satisfactorily addressed, some points need further clarification. The numberings are the same as in the original review.

We are pleased that the Reviewer is satisfied with our responses to their original concerns. We welcome their critical and perceptive suggestions to further refine our conclusions.

1. Phenotypic outcomes of essential gene KO are meaningful if clinical associations exist. Are there any clinical circumstances where SRSF1 loss (not upregulation) leads to liver diseases (not random cancer)? If the authors cannot spot global trends in GWAS or TCGA data, they can at least find and mention a number of relevant case studies (if any) in the introduction.

We are not aware of any previous reports demonstrating a direct link between SRSF1 loss and human liver disease. As indicated in our previous response, we did not find significant SRSF1 mutations in diseased human livers from the TCGA or GWAS datasets. However, it is possible that under certain circumstances (e.g., after hepatocellular injury, exposure to xenobiotics, or hepatotoxins), SRSF1 expression/activity within human hepatocytes is diminished, and that could have clinical relevance in the development of fatty liver disease. We have added this point to the Discussion section of the revised manuscript (**Lines 574-577**).

2. I think the author's rebuttal here is rather misleading. The focus should be on "how many and which genes" (not the SRSF1 binding sites nor the fractions) are incorrectly spliced upon SRSF1 KO. Can the authors provide the number of genes with splicing defects and their GO analysis results in the supplementary figures?

We apologize that the Reviewer missed these data in the original manuscript. We actually performed an in-depth analysis of alternative splicing on purified hepatocyte RNAs isolated from wildtype as well as SRSF1 chronic (10-day and 1-month timepoints) and acute (2-week) knockouts. These data can be found in **Supplementary Fig. 3a-d, Figure 7a, and Supplementary Fig. 5a, 5c**.

For instance, the *numbers* of differentially spliced regions ($|\Delta\text{PSI}| > 15\%$) organized according to the *splicing type* along with *increased vs. decreased* inclusion in the 10-day and 1-month-old SRSF1 HKO hepatocytes were provided in **Supplementary Fig. 3a**. The *distribution* of ΔPSI values for all differentially spliced regions from 10-day and 1-month timepoints were also shown in **Supplementary Fig. 3b**. Additionally, *Gene Ontology analysis* of transcripts with differentially spliced regions from 10-day and 1-month timepoints was presented in **Supplementary Fig. 3c**. Likewise, the *numbers, types* of splicing events, their ΔPSI *distributions* and integrated *Gene Ontology analysis* after acute depletion of SRSF1 were presented in **Supplementary Fig. 5a, 5c and the main Figure 7a**.

If the numbers reach hundreds and the genes have DNA damage-related roles, it would be rather hasty to rule out the splicing effects as 'secondary': In this case, the authors should at least clearly describe this limitation in the discussion section.

We respectfully disagree with the Reviewer about this remark. The number of misspliced genes is indeed in the hundreds, and in some cases, they might even have roles in "DNA damage response." However, in our view, these changes are secondary because they manifest a "response" to the widespread genotoxic stress, which is triggered by the undue formation of deleterious R-loops in SRSF1-deficient hepatocytes. Although we cannot rule out that some misspliced events exacerbate the hepatocellular damage in SRSF1-deficient hepatocytes, we believe they are not the primary contributing factor here. The DNA damage caused by the accumulation of R-loops is the main driving force for hepatocyte damage, resulting in many downstream effects. We have discussed these points and described the chain of events that led to the demise of SRSF1-deficient hepatocytes in the revised Discussion section of the manuscript (**Lines 587-591**).

3. I thought this point, i.e., 'R-Loop Induced DNA Damage' was the main argument of this paper, as best represented in the title "Splicing Factor SRSF1 Deficiency in the Liver Triggers NASH-like

Pathology *via R-Loop Induced DNA Damage and Cell Death*". If the authors are reluctant to address this point in the current manuscript, I suggest the authors remove the italics part from the title and tone down the respective statements throughout the manuscript.

As suggested by the Reviewer, we have removed the italicized part "*via R-Loop induced DNA damage*" from the title and made changes to tone down respective statements in the abstract and manuscript.

Reviewer #3 (Remarks to the Author):

The authors have made a number of changes to the manuscript that supports their conclusions. There are still a few issues that I think need to be addressed.

We thank the Reviewer for their continued in-depth evaluation of our manuscript. We welcome their insightful suggestions to improve the quality of this work further.

1. I thank the authors for including blinded scoring of the NASH histology. It raises some interesting questions. Firstly, the NAS scores need to be analyzed statistically before the authors can say that there is a difference in any score over time. These are ordinal scores, so normal t-tests cannot be used. A non-parametric test or chi-squared might work.

Thank you, as requested, we have now added the statistical analysis (non-parametric Mann-Whitney U Test, with p-values) to the blinded NAS histology scores in **Supplementary Table 2**.

Secondly, the authors state that inflammation, ballooning, and steatosis decrease, but fibrosis does not. Do the authors have an explanation of why fibrosis does not decrease?

We appreciate the Reviewer's curiosity regarding the lack of fibrosis reversal in SRSF1 HKOs. Indeed once considered an irreversible condition, recent studies have generated new excitement in the field that hepatic fibrosis can be reversed in some situations. In the case of SRSF1 HKOs, we do not have a clear explanation for why fibrosis is not fully resolved when steatosis, inflammation, and hepatocellular damage all normalize with age. We speculate that fibrotic changes in HKOs persist because (i) bridging intrahepatic fibrosis might have blocked or limited the flow of blood within the liver, which could have retarded or even prevented reversibility, (ii) widespread cell death and activation/recruitment of inflammatory cells following SRSF1 deletion might have led to a rapid period of collagen cross-linking, rendering the extracellular matrix less sensitive to degradation by enzymes over time, and/or (iii) the total content of collagen and other fibrogenic molecules, could have caused extensive scarring of the perisinusoidal spaces that became physically inaccessible or resistant to the degradative enzymes.

Also, the Supplementary Table is at odds with the histology shown in Fig. 3. The authors state that only slight bridging fibrosis remains at 6 months, but that's not what the histologist scores say.

The Reviewer raises a valid point regarding the discrepancy between the fibrosis seen by histology and the fibrosis scoring. Although histologically, the fibrosis appears to improve at 6 months when compared to 3 months, the scoring does not change significantly. The explanation for this inconsistency is the limitations of the scoring system to capture quantitative differences in fibrosis. The scoring of the fibrosis is primarily determined by the location and pattern of fibrosis. Therefore, while histologically, there is a reduction in fibrosis from 3 to 6 months, the pattern of fibrosis still remains, which is why the blinded scoring of the samples could not discriminate the improvement in fibrosis. We have modified our text in the revised Results section, where we clearly state that "*except for the intrahepatic fibrosis, the liver injury in SRSF1 HKOs subsided with age, and the mice recovered as SRSF1-expressing hepatocytes repopulated the liver*" (Lines 191-193).

2. It is difficult to compare transcription changes with changes in splicing. The transcriptional analysis sums all reads across a transcript, but splicing only looks at exon junction spanning reads. So the number of reads at each junction is very much lower than the number of reads per transcript. Indeed splicing junction data is very sparse. The number of events is much higher too, up to 10-20 fold more splicing events than genes. This has a big effect on the statistical analysis when multiple corrections is applied, as the n is much larger. It is often helpful to limit the analysis of splicing events to those

observed in all of, or the majority of, the samples. This reduces the n considerably. The analysis of splicing can give very different results when different approaches are used. The authors should analyze the data using a different approach from rMATS to see if they get the same results.

The Reviewer raises an important point: making direct numerical comparisons between transcriptional and splicing changes can sometimes be misleading, especially if the sequencing data is shallow. However, in our study, we deep sequenced poly(A) selected RNAs prepared freshly from hepatocytes isolated from wildtype and SRSF1-deficient livers, generating high-resolution transcriptome datasets with an average of >100,000 paired-end 100bp reads. Moreover, to increase the confidence of our analysis, we used the well-cited and one of the most popular alternative splicing analysis pipeline “rMATS version 3.2.5”, which filters data based on junctional read counts, taking into account the replicate data and the overall abundance of each transcript.

Over the years, we have had great success validating rMATS results with splicing-sensitive RT-PCRs [Bhate et al. (2015) *Nat Commun.*; Bangru et al. (2018) *Nat Struct Mol Biol.*]. Indeed, during the previous round of revision, we interrogated multiple genes from the rMATS output of RNA-seq data from wildtype and acSRSF1 HKO hepatocytes to represent splicing events increasing and decreasing in inclusion. As is shown in **Supplementary Figs. 6a, 6b**, the rMATS estimation of Δ PSI values correlated strongly with the Δ PSI values obtained by RT-PCR.

Nonetheless, based on the Reviewer’s suggestion, we performed additional alternative splicing analysis using an independent pipeline, Quantas [J. *Neurosci* 2014; 34 (36) 11929-11947; *Nat Commun.* 2018; 9: 2189]. We noted significant overlap in the *number*, *type*, and *magnitude* of splicing changes between rMATS and Quantas. These comparisons are presented in **Response Figure 1** below.

Response Figure 1: Differential splicing analysis in Acute SRSF1 HKO by rMATS and Quantas. (a) Venn diagrams showing the number of differentially spliced exons (FDR < 0.10, junction read counts ≥ 10 , and difference in Percent Spliced Index $|\Delta$ PSI| > 15%) by event type determined by rMATS and Quantas. E, skipped exon; MXE, mutually exclusive exons; RI, retained intron; A5SS, alternative 5’ splice site; A3SS, alternative 3’ splice site. (b) Bar plot showing percent overlap of differentially spliced exons determined by rMATS and Quantas by event type. Percent overlap was determined with respect to the number of events identified by Quantas. (c) Scatter plot showing the distribution of Δ PSI values for differentially spliced exons by event type between Quantas and rMATS.

3. Gene ontology analysis with alternative splicing is also tricky as, in most cases, it is not known what functional effect the events have (unlike transcription, where genes go up and down). So any conclusions from such an analysis should be taken with a grain of salt.

We totally agree with the Reviewer. With a splicing-centric research focus of our lab, we are keenly aware of the limitations of Gene Ontology analysis for alternatively spliced mRNAs. We recognize that assigning molecular function(s) to alternatively spliced regions is not trivial, as variable regions may affect the structure, activity, localization, stability, and/or binding interactions of the encoded proteins. That is why in the manuscript, we have not made *any* predictions or assertions about how the missplicing of individual transcripts might directly or indirectly contribute to the observed phenotypes of SRSF1 HKOs. Our main purpose in presenting the Gene Ontology data in **Supplementary Fig. 3 and Figure 7** is to provide general information to the reader who might be interested in knowing the gene categories that represent splicing defects in SRSF1-deficient cells.

4. The authors included cell cycle analysis as requested and show that siSRSF1 leads to G2/M arrest. Leading to the idea that DNA damage is causing arrest. It would be nice to show that the DNA damage is detected in the arrested cells by co-staining.

As the Reviewer correctly noted, we added the cell cycle analysis in our last round of revision and showed that SRSF1 deficiency leads to a G2/M arrest. Arrest at the G2/M phase is indicative of DNA damage accumulating during replication that is beyond repair. Although co-staining is one way to show that DNA damage is present in the arrested cells, in our view, a similar inference can be drawn from other experiments. For instance, we have independently demonstrated that SRSF1 depletion in primary mouse hepatocytes and human cell lines results in (a) extensive DNA damage as evidenced by γ -H2A.x staining (**Fig. 7b, 9a**), (b) induced expression and phosphorylation of p53 (**Supplementary Fig. 10d-e**), which is known to block cells at the G2 checkpoint through its inhibition of Cdc2, the cyclin-dependent kinase required to enter mitosis, and (c) the induction of many known p53-responsive genes (**Supplementary Fig. 10c**). Moreover, in our siRNA experiments, over 90% of SRSF1-deficient cells showed positive staining for the DNA damage marker γ -H2A.x whereas FACS analysis demonstrated nearly 64% of SRSF1-depleted cells to be arrested in the G2/M phase (**Fig. 9**). Collectively, these data imply that widespread DNA damage triggered due to the accumulation of R-loops is the most likely reason leading to the G2/M arrest of SRSF1-deficient cells.

5. I think the mechanistic question is still unanswered. How does loss of SRSF1 lead to increased R-loops? Is it related to the binding of SRSF1 to the Pol 2CTD?

The Reviewer is correct that, at this point, we cannot fully explain how SRSF1 deficiency leads to the accumulation of R-loops; and whether it is directly related to the binding of SRSF1 to the c-terminal domain of RNA Pol-II. These lines of investigation are logical extensions of this work, and we will be investigating them in the future, but they fall outside the scope of the current study.